# Preventing excessive autophagy protects from the pathology of mtDNA mutations in *Drosophila melanogaster*

Najla El Fissi[1], Florian A. Rosenberger [2,8], Kai Chang[1,8], Alissa Wilhalm[1], Tom Barton-Owen[3,4], Fynn M. Hansen[2], Zoe Golder[3,4], David Alsina [1,5], Anna Wedell[5,6], Matthias Mann [2,7], Patrick F. Chinnery[3,4], Christoph Freyer [1,5,9] ✉ & Anna Wredenberg [1,5,9] ✉

Aberration of mitochondrial function is a shared feature of many human pathologies, characterised by changes in metabolic flux, cellular energetics, morphology, composition, and dynamics of the mitochondrial network. While some of these changes serve as compensatory mechanisms to maintain cellular homeostasis, their chronic activation can permanently affect cellular metabolism and signalling, ultimately impairing cell function. Here, we use a *Drosophila melanogaster* model expressing a proofreading-deficient mtDNA polymerase (POLγ^exo·) in a genetic screen to find genes that mitigate the harmful accumulation of mtDNA mutations. We identify critical pathways associated with nutrient sensing, insulin signalling, mitochondrial protein import, and autophagy that can rescue the lethal phenotype of the POLγ^exo· flies. Rescued flies, hemizygous for *dilp1*, *atg2*, *tim14* or *melted*, normalise their autophagic flux and proteasome function and adapt their metabolism. Mutation frequencies remain high with the exception of *melted*-rescued flies, suggesting that *melted* may act early in development. Treating POLγ^exo· larvae with the autophagy activator rapamycin aggravates their lethal phenotype, highlighting that excessive autophagy can significantly contribute to the pathophysiology of mitochondrial diseases. Moreover, we show that the nucleation process of autophagy is a critical target for intervention.

Pathogenic mutations within the mitochondrial genome (mtDNA) are known to cause a range of rare inherited diseases, collectively referred to as mitochondrial diseases[1,2]. Additionally, mtDNA variants are linked to numerous conditions, including neurodegeneration, heart disease, autoimmune disorders, as well as psychiatric illnesses, and are believed to play a role in shaping numerous complex traits[3–10].

The accumulation and expansion of both de novo and inherited mtDNA mutations are associated with age-related declines in mito-

[1]Department of Medical Biochemistry and Biophysics, Karolinska Institutet, 171 65 Stockholm, Sweden. [2]Proteomics and Signal Transduction, Max-Planck Institute of Biochemistry, Martinsried 82152, Germany. [3]Department of Clinical Neurosciences, School of Clinical Medicine, University of Cambridge, Cambridge Biomedical Campus, Cambridge, UK. [4]Medical Research Council Mitochondrial Biology Unit, Cambridge Biomedical Campus, Cambridge, UK. [5]Centre for Inherited Metabolic Diseases, Karolinska University Hospital, 171 76, Stockholm, Sweden. [6]Department of Molecular Medicine and Surgery, Karolinska Institutet, 171 65 Stockholm, Sweden. [7]Faculty of Health Sciences, NNF Centre for Protein Research, University of Copenhagen, Copenhagen 2200, Denmark. [8]These authors contributed equally: Florian A. Rosenberger, Kai Chang. [11]These authors jointly supervised this work: Christoph Freyer, Anna Wredenberg. ✉ e-mail: christoph.freyer@ki.se; anna.wredenberg@ki.se

chondrial content and function, leading to biochemically compromised cells, which has been associated with the natural aging process[11–16]. Given that mitochondria have a limited DNA repair system, it is crucial to identify other mechanisms that maintain mitochondrial integrity, enhance biogenesis, selectively remove dysfunctional mitochondria, or prevent deleterious mtDNA mutations[17–19].

We and others previously tested the impact of accumulating mtDNA variants in disease, for instance, by expressing a mitochondrially-targeted cytidine deaminase in flies[20] or a proof-reading deficient mtDNA polymerase gamma (POLγ$^{exo-}$, mtDNA mutator) in yeast, worms, flies, and mice[21–26]. These models accumulate mtDNA point mutations at an accelerated rate, resulting in progressive mitochondrial dysfunction and phenotypes characteristic of both mitochondrial diseases and the ageing process. The variability in the phenotypes of the different mtDNA mutator models is particularly striking in the fly, with viability depending on the type and amount of mutational burden in the mitochondrial genome[20,21,26].

In this work, we postulate that the progressive mitochondrial dysfunction observed in these mtDNA mutator models provides the opportunity to identify mechanisms that enhance tolerance to mtDNA mutations, reduce the mutation burden, or compensate for the OXPHOS dysfunction. By performing a genetic deficiency screen of chromosome 3 in *Drosophila melanogaster*, we identify several pathways associated with nutrient sensing and autophagy that upon heterozygous deletion prevent the excessive turnover of mitochondria in response to the accumulation of mtDNA mutations. Furthermore, we show that the autophagy activator rapamycin can aggravate the mtDNA mutator phenotype, while chemically inhibiting autophagy normalised mitochondrial function and rescued homozygous mtDNA mutator flies.

## Results

### A genetic screen in mtDNA mutator flies

We previously demonstrated that analogous to other proof-reading deficient mtDNA POLγ $^-$models, homozygous POLγ$^{exo-}$ flies progressively accumulate mtDNA deletions and point mutations, but unlike their murine equivalent, die at the third instar larvae stage[21]. To reveal cellular pathways that directly respond to this mitochondrial dysfunction, we performed a comprehensive screening of the entire third chromosome of the fruit fly genome to identify genes that, upon copy number reduction, can rescue the larvae lethal phenotype of homozygous POLγ$^{exo-}$ mtDNA mutator flies (Fig. 1a). The POLγ$^{exo-}$ mtDNA mutator flies were previously generated by ´ends-out´ homologous recombination, replacing the wild-type *tamas* locus (*Tamas* encodes for POLγ in *Drosophila*) with a control (*tamas*) or a D263A (POLγ$^{exo-}$) mutant genomic fragment to ensure physiological expression (see materials and methods)[21].

mtDNA mutator flies were crossed to 242 isogenic hemizygous deficiency strains (Dfs) from the DrosDel project[27] to form flies homozygous for the POLγ$^{exo-}$ allele (mtDNA mutator) and hemizygous for overlapping genomic regions of chromosome 3 (Supplementary Fig. 1, Supplementary Data 1). In this way, we collectively screened 6380 genes, equivalent to ~40% of the fly genome. We identified 17 loci that, upon heterozygous deletion, resulted in the emergence of viable homozygous adult POLγ$^{exo-}$ flies. Using an additional 159 individual gene and p-element insertion mutants (Supplementary Data 1), we identified nine genes that, in a heterozygous state, rescued homozygous POLγ$^{exo-}$ mutator flies (Fig. 1b, c, Supplementary Data 1). These flies (hereafter termed rescued flies) were homozygous for the POLγ$^{exo-}$ allele (*tamas*$^{D263A/D263A}$) and showed a reduced expression of the relevant rescue gene compared to homozygous mtDNA mutator flies (Supplementary Fig. 2a–d).

The identified rescue genes can be classified into autophagosome formation (*atg2*, *atg6*, *atg14*, and *cg11975*)[28], mitochondrial protein import (*tim14*)[29], insulin-like growth signalling (IIS) (*dilp1*,

*inR*, and *glyS*)[30–32], and nutrient sensing (*melted*)[33–35] (Fig. 1c). For an additional Dfs we reduced the responsible locus to *tim17a2*, *tim17b1*, or *cg31538*. Tim17a2, Tim17b1, and Tim14 are all part of the mitochondrial pre-sequence translocase Tim23, regulating mitochondrial matrix protein import[29], while *cg31538* encodes a protein of unknown function associated with muscle development. It is thus possible that all three components of the Tim23 complex can rescue the mtDNA phenotype. Heterozygous KO of *tom40*, a member of mitochondrial outer membrane import machinery, on the other hand, did not rescue the mtDNA mutator line (Supplementary Fig. 3a). The remaining seven Dfs lacked suitable mutant fly models, preventing us from identifying the responsible genes (Supplementary Data 1). Our genetic screen uncovered pathways directly impacting mtDNA mutation-related pathology, emphasising their modulation as a potential therapeutic approach to attenuate mitochondrial dysfunction.

### Nutrient sensing affects the mtDNA mutator larvae

Several identified factors are involved in nutrient sensing, such as the glycogen synthase *glyS*, the insulin receptor *inR*, and the *Drosophila* insulin-like peptide, *dilp1*. Additionally, *melted* (*melt*), whose deletion also resulted in adult mutator flies, is believed to be part of the insulin/PI3K signalling pathway by modulating the activities of the forkhead box protein FOXO and TOR[33–35], suggesting an overall involvement of glucose availability in the rescue mechanism. In agreement, the deletion of *dfoxo*, a negative effector of the IIS pathway, not only failed to rescue mtDNA mutator flies but aggravated the mutator phenotype by being lethal as double heterozygous (+/POLγ$^{exo-}$; +/*dfoxo*$^{PE}$) larvae (Supplementary Data 1).

*Drosophila* has an inverse receptor-ligand configuration compared to mammals and expresses eight different *Drosophila* insulin-like peptides (Dilps) in a spatiotemporal pattern that act via InR[30,32,36,37]. Several Dilps have been reported to complement one another, and we, therefore, crossed the POLγ$^{exo-}$ allele into mutants of *dilp2* and *dilp3*, as well as multi-mutant lines hemizygous for *dilps(1,2,3,4,5)*, *dilps(1,2,3,4,5,6)*, *dilps(1,2,3,4,5,7)* or *dilps(2,3,5,7)*. However, only the heterozygous deletion of *dilp1* resulted in viable adults, leaving Dilp1 as the only ligand crucial for mutator lethality (Supplementary Fig. 3b, c,d).

### Restoration of OXPHOS activity in rescued larvae

We chose four rescue lines, heterozygous for *atg2* (BDSC#17156), *tim14* (BDSC#17682), *dilp1* (BDSC#78055), and *melted* (BDSC#19928) and homozygous for the POLγ$^{exo-}$ mutator allele (*tamas*$^{D263A/D263A}$; *rescue gene*$^{+/-}$; hereafter termed "rescue-gene"-mut), for further characterisation. No apparent differences in hatching rates were noted, with adults living for at least ten days after eclosure when we stopped observing the animals. However, one-day-old rescue flies exhibited a reduced climbing capacity compared to *tamas* controls (Fig. 1d). To dissect the rescue mechanisms, our investigation focused on the larvae stage, allowing us to compare rescued to mtDNA mutator larvae. Rescued larvae had a normalised size, and an almost fully restored mobility compared to mtDNA mutator larvae (Fig. 1e, Supplementary Movie 1). Moreover, the activities of the mitochondrial respiratory chain enzyme complexes I, I + III, II + III, IV and V were significantly improved, with *dilp1*- and *melt-mut* rescue larvae showing a complete normalisation (Fig. 1f). Western blot analysis confirmed that NDUFS3 steady-state levels returned to control levels upon rescue (Fig. 1g, h).

Next, we assessed the mitochondrial membrane potential in the larvae's middle midgut[38,39] using tetramethylrhodamine and ethyl ester (TMRE) staining (Fig. 1i, j and Supplementary Fig. 4). Homozygous mtDNA mutator larvae exhibited a significantly reduced mitochondrial membrane potential, which was partially restored in the *rescue-mut* larvae. Furthermore, aconitase activity, which is sensitive to reactive oxygen species (ROS), was decreased in the mutator and normalised in the rescue larvae (Fig. 1k). Together, our results show that the

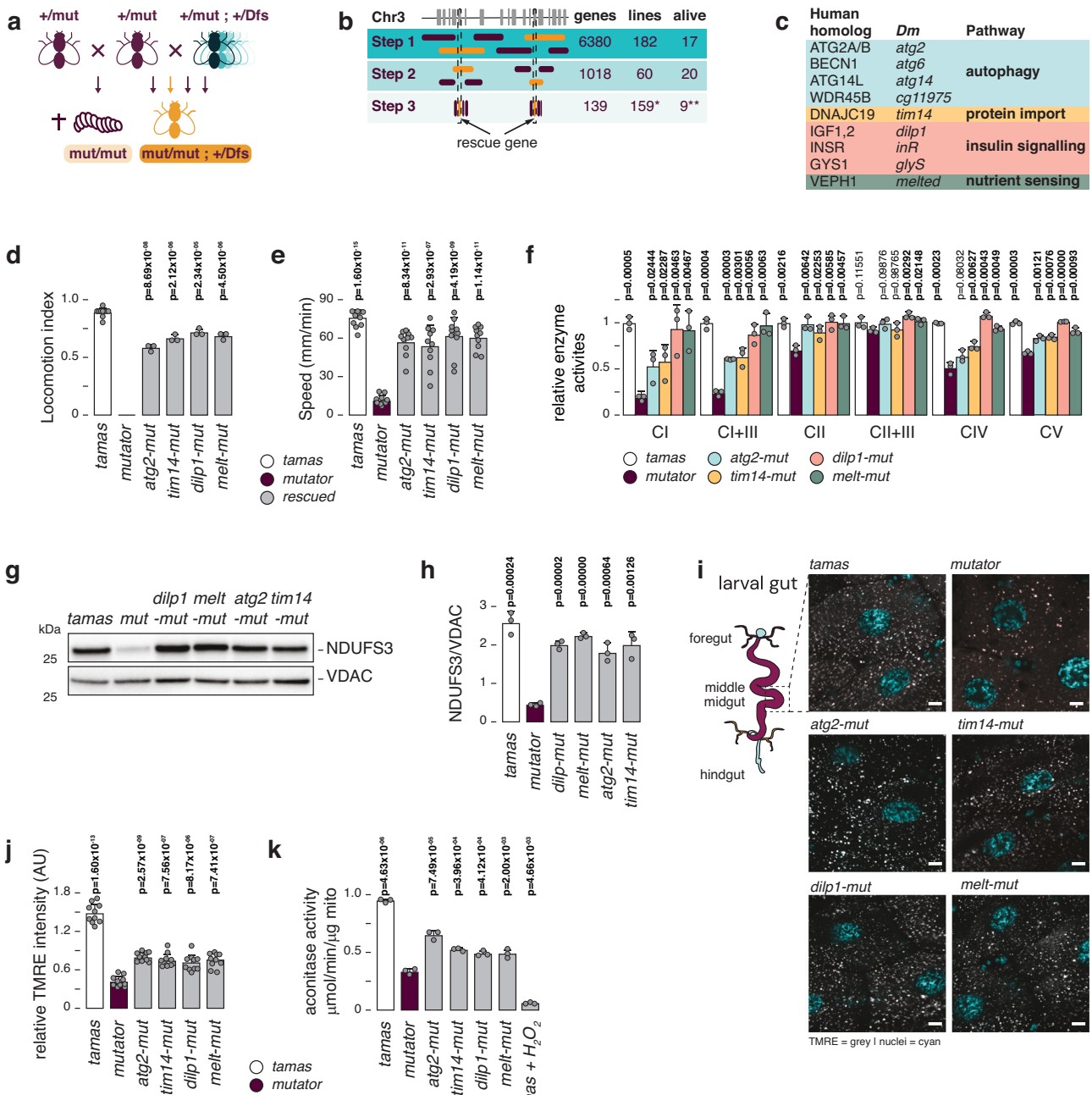

**Fig. 1 | A genetic deficiency screen in mtDNA mutator flies. a** Cartoon depicting the genetic screen strategy. Male heterozygous POLγ$^{exo-}$ ( + /mut) flies are crossed to a library of heterozygous deficiency strains (Dfs). Resulting double heterozygous (+/POLγ$^{exo-}$; +/Dfs) females are then selected for crosses with fresh heterozygous POLγ$^{exo-}$ ( + /mut) males to screen for viable homozygous POLγ$^{exo-}$ (mut/mut; +/Dfs) adult flies. **b** Deficiency strains were used in steps 1 and 2, followed by testing individual genes, using mutant and P-element insertion mutant fly lines (step 3). The number of genes tested (genes), number of fly strains (lines) used, and number of rescued lines (alive) are shown. **c** The identified mitorescue genes are shown, as well as their human homologs, and biological pathways. **d** Negative geotaxis test (NGT), measuring fly locomotion of *tamas* control (white), *mutator* (plum), and rescued (grey) flies 2 days after eclosure. Mean values of number of flies tested are shown. $N$ = 3 biological replicates with 10 flies per replica. For *tamas* controls $N$ = 10 biological replicates with 10 flies per replicates. **e** Crawling distance of third instar larvae from *tamas* control (white), *mutator* (plum), and rescued (grey) flies. Mean values of distance crawled per minute are shown. $N$ = 10 biological replicates per genotype. **f** Relative mitochondrial respiratory chain enzyme activities in isolated mitochondria from third instar larvae. Measurements were performed for NADH:ubiquinone oxidoreductase (CI), NADH:cytochrome c oxidoreductase (CI + III), succinate:ubiquinone oxidoreductase (CII), succinate:cytochrome c

oxidoreductase (CII + III), cytochrome c oxidase (CIV), and ATP synthase (CV). *Tamas* controls are shown in white, *mutator* in plum, *atg2-mut* in blue, *tim14-mut* in yellow, *dilp1-mut* in pink, and *melted-mut* in green. Mean values relative to *tamas* are shown. $N$ = 3 biologically independent samples per genotype. **g** Western blot analysis of protein extracts from third instar larvae, decorated with antibodies against the OXPHOS complex I subunit NDUFS3, and VDAC. **h** Quantification of (**g**). Men values of NDUFS3 relative to loading control (VDAC) are shown. *Tamas* controls are shown in white, *mutator* in plum, and rescued larvae in grey. $N$ = 3 biological replica. **i** Representative images of TMRE staining (grey) of gut tissue from third instar larvae. Nuclei are stained with Hoechst (cyan). Scale bar = 9 μm. **j** Quantification of (**i**). Mean values of percentage of relative image intensity are shown. $N$ = 10 images per genotype. Tamas controls are shown in white, mutator in plum, and rescued larvae in grey. **k** Aconitase activities in fresh mitochondria of third instar larvae. Mean values per μg of mitochondria are shown. *Tamas* controls are shown in white, *mutator* in plum, and rescued larvae in grey. $N$ = 3 biologically independent samples per genotype. Student's two-tailed *T*-test was used with mutators (*mut*) against other genotypes, except for (**d**) where values were compared to *tamas*. $P$ values < 0.05 are shown in bold. Error bars represent Standard deviation. Source data are provided in the main figure source data file.

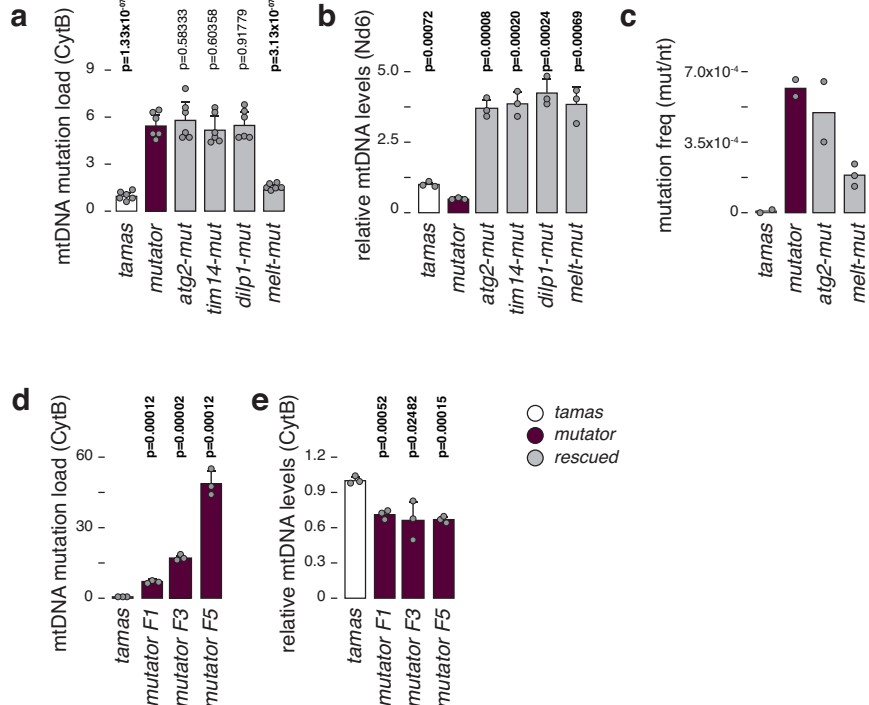

**Fig. 2 | mtDNA mutation load in mtDNA mutator and rescued flies. a** Relative mtDNA mutation load in third instar larvae, determined by random mutation capture assay (RMC). Mean values relative to *tamas* are shown. $N = 6$ biologically independent samples with 10 flies each per genotype and 3 technical replicates. **b** Relative mtDNA levels in third instar larvae, using ND6 as mtDNA and rp49 as nuclear targets. Mean values relative to *tamas* are shown. $N = 3$ biologically independent samples with 3 technical replicates for each genotype. **c** mtDNA mutation frequency determined by cloning and sequencing. Mean values per nucleotide sequenced are shown. For *tamas*, *mutator*, and *atg2-mutator* samples $N = 2$ biologically independent samples, with at least 180 clones sequenced per genotype were used. For *melt-mutator* $N = 3$ independent biological replicates were used.

**d** Relative mtDNA mutation load in third instar larvae, determined by random mutation capture assay (RMC). Mean values relative to *tamas* are shown. $N = 3$ biologically independent samples with 10 flies each per genotype and 3 technical replicates. **e** Relative mtDNA levels in third instar larvae, using Cytb as mtDNA and rp49 as nuclear targets. Mean values relative to *tamas* are shown. $N = 3$ biologically independent samples with 3 technical replicates for each genotype. *Tamas* controls are shown in white, *mutator* in plum, and rescued larvae in grey. Student's two-tailed T-test was used with mutators (*mut*) against other genotypes. *P* values < 0.05 are shown in bold. Error bars represent Standard deviation. Source data are provided in the main figure source data file.

heterozygous deletion of *atg2*, *tim14*, *dilp1* or *melted* rescues the impaired mitochondrial function of the homozygous mutator larvae.

## mtDNA mutation load determines survival

We previously reported that mtDNA mutator larvae have reduced mtDNA levels with increased unique mtDNA mutations ($\sim 2 \times 10^{-4}$ mutations/bp), characterised by a higher frequency of transitions versus transversions but equal codon distribution[21]. We performed random mutation capture (RMC) analysis at two different sites of the mitochondrial genome to estimate the mutational burden in the individual rescue lines (see materials and methods). We found no difference in mutation load between the mtDNA mutator and *atg2-*, *dilp1-* or *tim14*-rescue larvae. In contrast, *melt-mut* larvae kept a mutation burden comparable to controls (Fig. 2a, Supplementary Fig. 5a). mtDNA levels were increased 3-fold in the rescued lines, suggesting that increasing mtDNA copy number is part of the rescue mechanism (Fig. 2b, Supplementary Fig. 5b).

We confirmed the RMC results by cloning and sequencing an 825 bp mtDNA fragment across *atp6* and *cox3* of *tamas*, *mutator*, *atg2-mut*, and *melt-mut* larvae derived from at least two independent crosses (Fig. 2c, Supplementary Data 2). We sequenced at least 180 clones from each cross ($\sim 1.5 \times 10^5$ base pairs per cross) and aligned them to the *wDah* mtDNA genome (KP843845.1; position 4100 to 4924). *Tamas* control larvae exhibited a total mutation frequency of $6.49 \times 10^{-06}$ per base pair, while homozygous mutator (POLγ$^{D263A/D263A}$) larvae showed an increased total frequency of $6.17 \times 10^{-04}$, reflecting previous

results[21,40]. In agreement with the RMC results, the mutation frequency was high in *atg2-mut* larvae ($4.94 \times 10^{-4}$) and reduced in *melt-mut* samples ($1.85 \times 10^{-4}$), albeit not as low as control larvae. Interestingly, *melt-mut* larvae retained a relatively high frequency of unique mutations but still substantially lower than mtDNA *mutator* larvae (Supplementary Data 2). Together, these results suggest a more frequent clonal expansion of mtDNA mutations in the *mutator* larvae.

Next, we attempted to rescue larvae with increased mutational burden by intercrossing heterozygous POLγ$^{exo-}$ flies for 1, 3 or 5 generations before targeting the rescue genes. As expected, the mutation load increased in homozygous mutator larvae with consecutive intercrosses, consistent with previous reports[26,41]. While the mtDNA copy number remained consistent throughout the generations, only the F1 *rescue-mut* generation could be rescued (Fig. 2d, e). Together, this supports that the accumulation of mtDNA mutations drives the phenotype in mutator larvae and that the rescue potential in the homozygous POLγ$^{exo-}$ flies is dependent on the mutational burden in the female germline.

## Proteomic analysis reveals partial restoration of OXPHOS subunits

To better understand the cellular processes involved in the mutator phenotype and its rescue, we performed mass-spectrometry-based proteomics from larvae, fat body, and brain samples from mtDNA mutator, rescue, and control lines (see materials and methods, Supplementary Data 3). Principal component analysis (PCA) presented a

distinct separation between the individual genotypes, with control samples separating from the other genotypes (Fig. 3a, Supplementary Fig. 6a–h). Global comparison between mutator and rescued samples showed tissue-specific changes, with the brain mainly adjusting its glycan metabolism, while the category amino acids was prominent in total larvae samples (Supplementary Fig. 7a). Pathways related to catabolic processes of intermediary metabolism were significantly reduced in mtDNA mutator samples and reversed again in the rescued lines.

Notably, pathways including OXPHOS, glycolysis, the TCA cycle, and glutathione metabolism recovered in the rescue samples (Fig. 3b, c, Supplementary Fig. 7b, c, 8, 9). Likewise, categories, such as DNA repair or glycosaminoglycan degradation, were increased in mutator samples but normalised again to control levels in the rescued lines.

Interestingly, the rescue samples presented notable differences in several pathways, such as lipid metabolism and branched-chain amino acid degradation, which significantly increased compared to control and mutator samples. Additionally, the downregulation of the proteasome was specific for the rescue larvae and particularly evident in the fat body (Fig. 3d, Supplementary Fig. 8), indicating that the reduced protein turnover alleviates mitochondrial stress. A marked increase in the category of mitochondrial translation accompanied this in the rescued samples (Fig. 3e, Supplementary Fig. 7b, c). Notably, the proteomic profiles were unique to the *rescue-mutator* lines as the individual heterozygous or homozygous inactivation of *atg2*, *tim14*, *dilp1*, and *melt* alone showed different proteomic changes (Supplementary Data 4). PCA revealed only mild clustering of the various genotypes, with homozygous *melted* and heterozygous *dilp1* KOs showing the most variation (Supplementary Fig. 10a). Furthermore, factors involved in OXPHOS showed only mild changes, with heterozygous and homozygous *melted* larvae exhibiting a marked downregulation of complex I subunits, contrasting observations made in the rescued larvae (Supplementary Fig. 10a).

## Reduced mitochondrial mass in mutator larvae
The observed proteomic changes and the increased mtDNA copy number are characteristic of an attempt to reduce mitochondrial turnover or increase the de novo biogenesis. Indeed, we observed a distinct decrease of mitochondrial summed protein intensities by mass spectrometry in mutator larvae, which normalised to and even increased beyond controls in the rescued larvae (Fig. 4a). In agreement, immunostaining of larvae brains showed a reduced mitochondrial mass in mutator samples, which increased again in the rescued samples (Fig. 4b, c). This could indicate that the mitochondria-specific removal, known as mitophagy, is upregulated in the mutator larvae. Mitochondrial morphology was severely affected in mutator larvae, preventing us from obtaining reliable measurements (Supplementary Fig. 11a). We, therefore, first inhibited mitochondrial turnover using the autophagosome-lysosome fusion inhibitor bafilomycin A1 (BafA1), followed by confocal imaging to investigate mitochondrial morphology in the fat bodies of control (*tamas*), *mutator*, and *rescue* larvae (Supplementary Fig. 11b–d, e). While the number of branches, branch length, and junctions significantly increased in the mutators, the mitochondrial morphology partially normalised again in *rescue-mut* larvae.

Our proteomic data revealed no significant alterations in mitophagy-specific factors, except for a notable reduction in the mitophagy receptor BCL2 interacting Protein 3 (BNIP3)[42] in *melted-mut* larvae (Fig. 4d). Interestingly, transcript levels of the peroxisome proliferator-activated receptor gamma coactivator 1-alpha (PGC-1α), a potent inducer of mitochondrial biogenesis encoded by *spargel* in *Drosophila*, was only upregulated in *atg2*- and *dilp1*-rescue larvae and remained close to control levels in *tim14*- and *melted*- rescue larvae[43] (Supplementary Fig. 12a). Together these results indicate that

increased mitochondrial biogenesis or targeted mitophagy is not part of the overall rescue mechanism.

However, in addition to a reduction of mitochondrial content, we also observed a marked down-regulation of several peroxisomal proteins in the mutator larvae (Fig. 4e, Supplementary Fig. 8). Together, this reflects the activation of macroautophagy in the mutator larvae and although factors associated with macroautophagy were not strongly represented in our proteomic analysis, mutator larvae had a marked upregulation of the autophagic process on transcript level (Supplementary Fig. 12b). Furthermore, transmission electron microscopy (TEM) on muscle tissue revealed a disorganised muscle structure and the presence of large vacuoles in close contact to mitochondria, and even engulfing mitochondria in mutator larvae, suggestive of increased lysosome formation. In contrast, all four rescue lines showed a restructured muscle morphology and an apparent reduction in these intracellular vacuoles (Fig. 4f).

## Increased autophagy in mutator larvae
Next, we stained the brain and midgut with lysotracker, revealing increased lysosomal foci in mutator samples (Fig. 5a, b and Supplementary Fig. 12c). This increase in lysosomal foci normalised again in the rescued larvae, a trend supported by a general decrease of lysosomal proteins in rescued larvae, fat body and brain samples (Fig. 5c, Supplementary Fig. 9). Cathepsin B activity, an indicator of lysosomal function, showed no significant difference between controls, mutator, and *rescue-mut* larvae in gut tissue (Supplementary Fig. 12d, e). This suggests that despite an increase in lysosome number, lysosome activity per cell is unaffected, pointing to an increased autophagic flux in the mutator larvae, which normalises again in the rescued larvae.

Inhibiting autophagosome-lysosome fusion with BafA1 resulted in a robust translocation of Atg8 from the nucleus to the cytosol in mutator samples, which was much less pronounced in rescued samples (Fig. 5d). Co-staining for mitochondria exhibited an increased colocalisation between autolysosomes and mitochondria in the mutator samples, strengthening the idea of increased autophagic activity in the mutator larvae (Fig. 5e). Increased levels of lipidated Atg8 after BafA1 treatment confirmed an elevated autophagic activity in the mutator larvae (Supplementary Fig. 12f, g). Together, our results suggest that a higher mutation rate in the mtDNA mutator larvae leads to metabolic adaptations and increased autophagic flux, ultimately contributing to the pathophysiology seen in the mutator larvae.

We tested this by growing control and mutator larvae on rapamycin, an autophagy activator, by inhibiting TOR. Interestingly, mutator and control larvae were both affected by the treatment, with mutator larvae not developing past the second larval stage and control larvae presenting with a reduced pupation and hatching rate (Fig. 5f). This was accompanied by reduced mtDNA steady state and NDUFS3 protein levels, as well as an increased number of lysosomal foci (Fig. 5g, h and Supplementary Fig. 13a, b). These results strongly support the idea that an increased mitochondrial turnover can harm the organism. However, the exact mechanism remains unclear, as the deletion of other factors involved in TOR signalling, such as the TOR complex 1 and 2 subunits LST8, the Telo2 interacting protein 1(Tti1), and the rapamycin-insensitive companion of Tor (Rictor), as well as the TSC complex subunit 2 (TSC2), the serine/threonine kinase, LKB1 and AMP kinase (AMPK) did not affect mutator survival (Supplementary Fig. 3b and 13c–h).

## Nucleation process rescues the mutator lethality
Notably, our screen did not identify any mitophagy factors explicitly involved in clearing damaged mitochondria to improve cell function[17,18,44]. For instance, heterozygous inactivation of the E3 ubiquitin ligase Parkin, which protects against stress-induced mitochondrial dysfunction, did not lead to fly survival, suggesting that reducing mitochondrial turnover alone is insufficient to rescue the flies. Thus,

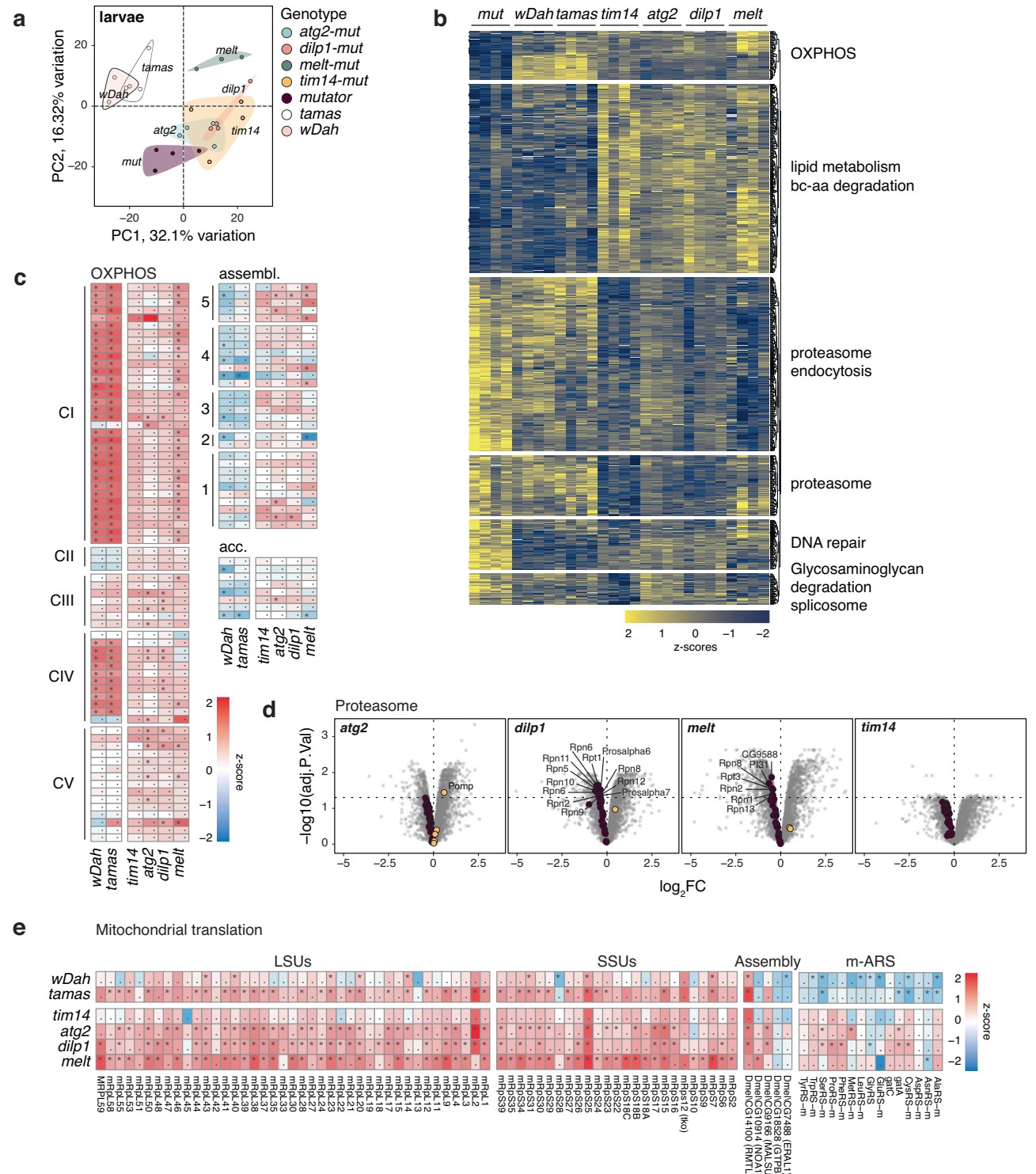

**Fig. 3 | Normalisation of the fly larvae proteome upon rescue. a** Principal components 1 and 2 of larval proteomes. Each dot is one sample. Colour overlay groups genotypes. **b** Expression of significantly different proteins (ANOVA, FDR < 0.01) as z-scores. Functional categories are significantly enriched gene ontologies (Fisher's exact test, FDR < 0.05). Each column is one replicate. **c** LogFC of OXPHOS, assembly and accessory subunits relative to mutator larvae and pooled by genotype (*n* = 4). **d** Volcano plots of respective genotypes against mutator larvae proteomes with the proteasome category highlighted. **e** LogFC of proteins involved in mitochondrial translation relative to mutator larvae and pooled by genotype (*n* = 4). * indicates significant hits (multiple-testing adjusted *p*-values, FDR < 0.05).

the mitochondrial dysfunction likely triggers additional cellular adaptations, such as proteasome activation or changes to amino acid metabolism, which alter cell function. To test whether disruption of autophagy per se can rescue the mutator larvae, we also targeted factors involved in autophagy initiation (*atg1*, *atg13*, *FIP700*), additional factors of nucleation (*atg18a*, *vps15*), elongation (*atg3*, *atg4b*, *atg8b*), autophagosome formation (*atg5*, *atg12*, *atg16*), or fusion with the lysosome (*syx17*, *nSyb*), by heterozygous deletion in the mtDNA

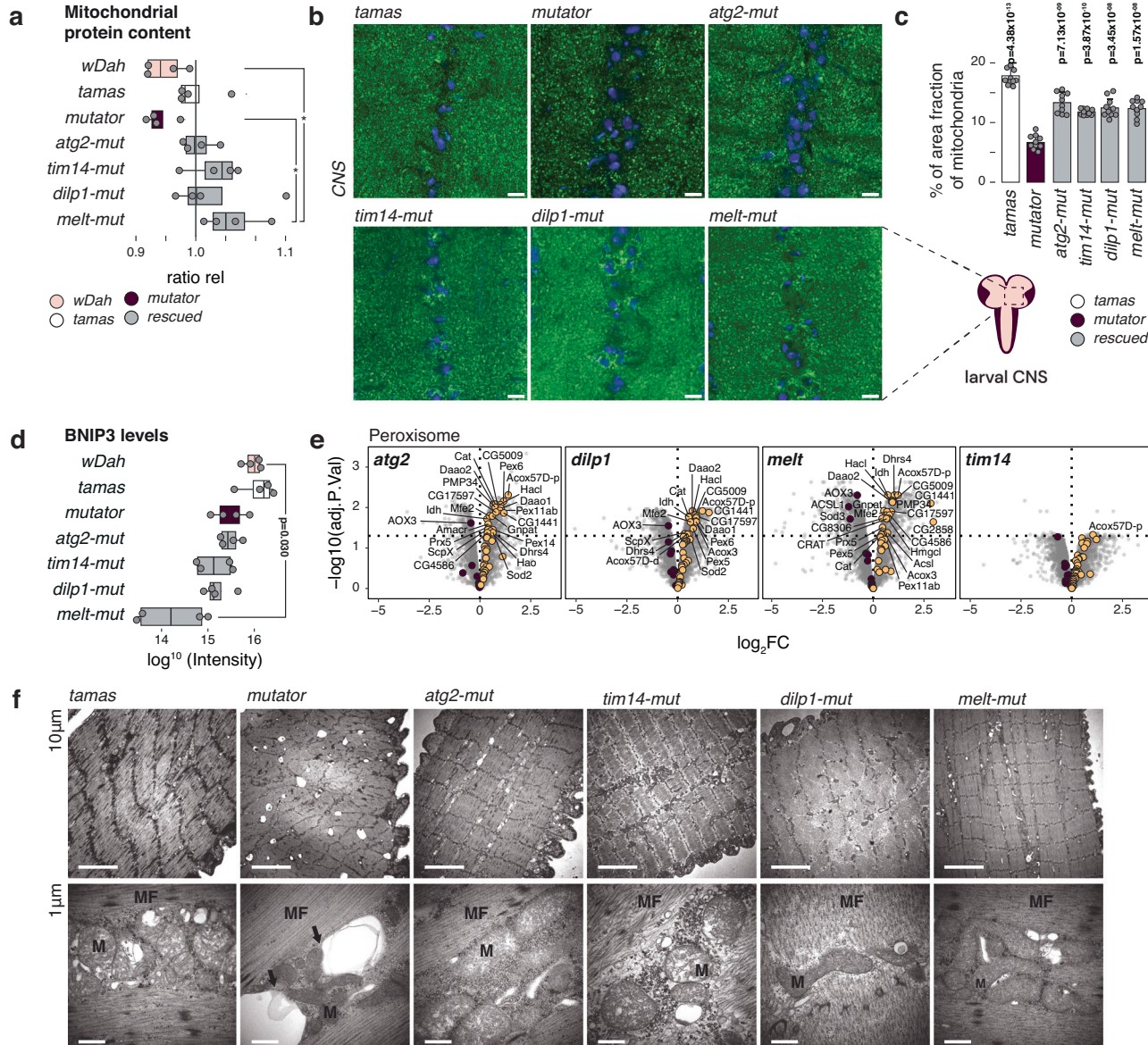

**Fig. 4 | Reduced mitochondrial mass in mtDNA mutator larvae. a** Relative abundance of the mitochondrial proteome normalised to the summed mean protein intensities in *tamas*. Data is presented as the larvae intensity ratio of mitochondrial proteins relative to *tamas* (ratio rel). *wDah* (pink), *tamas* (white) controls, *mutator* (plum), and rescued (grey) larvae are shown. $N = 4$ independent biological replicates. Boxplots represent the first and third quartile with median indicated. Whiskers represent the ±1.5 interquartile range. **b** Confocal images of the larval ventral nerve chord showing mitochondria in green (α-ATP-synthase) and nucleus in blue (DAPI). (Scale bar = 6 μm; Zoom = 1.8X). **c** Quantification of (**b**) with *tamas* (white) controls, *mutator* (plum), and rescued larvae (grey). Mean percentage of area fraction of mitochondria analysed. The average of ten independent images per genotype is shown. Student's two-tailed *T*-test was used with mutators (*mut*) against other genotypes. *P* values < 0.05 are shown in bold. Error bars represent Standard deviation. $N = 10$ biological replicates per genotype. **d** Abundance of

BNIP3 related peptides is shown for the genotypes as indicated. $N = 4$ independent biological replicates. Boxplots represent the first and third quartile with median indicated. Whiskers represent the ±1.5 interquartile range. *wDah* (pink), *tamas* (white) controls, *mutator* (plum), and rescued (grey) larvae are shown. **e** Volcano plots of respective genotypes against mutator larvae proteomes with the peroxisome category highlighted in plum (down) and yellow (up). Only rescued samples are shown. Control samples are shown in the supplementary files. **f** Transmission electron microscopy sections of third-instar larvae muscle. Top panels: section showing double membrane vesicular structures evocating autophagy figures in *mutator* larvae. These structures sometimes contact (*mutator*, arrows), or contain mitochondria (*mutator*, arrows). Bottom panels: higher magnification views. (arrows = autophagy like vesicle, M=mitochondria, MF= muscle fibre). Representative images from $N = 3$ biological replicates (20 images per replicate) are shown.

mutator flies. Surprisingly, only the heterozygous deletion of the nucleation-associated factors *atg18a* or *vps15* resulted in viable adult mtDNA mutator flies (Fig. 6a, Supplementary Data 1). Heterozygous deletion of six of 17 autophagy-related factors tested rescued the mutator larvae, with all six being part of the nucleation process[28,45,46]. We tested this by growing control and homozygous POLγ[exo-] larvae in the presence of inhibitors specific to autophagy initiation (KU-55933), nucleation (3-methyladenine), or autolysosome formation (BafA1) at

three different concentrations. Since treatment during the pupae stage is difficult and untreated homozygous mtDNA mutator larvae do not develop past the 3rd instar larvae stage, we used the pupation rate as treatment readout. In support of our genetic findings, larvae grown in the presence of 3-methyladenine (3-MA) developed to the pupae stage at all concentrations (Fig. 6b). Interestingly, KU55993 and BafA1 also resulted in a partial rescue, though to a lesser extent. Notably, only 3-MA improved larvae motility and decreased lysosomal foci

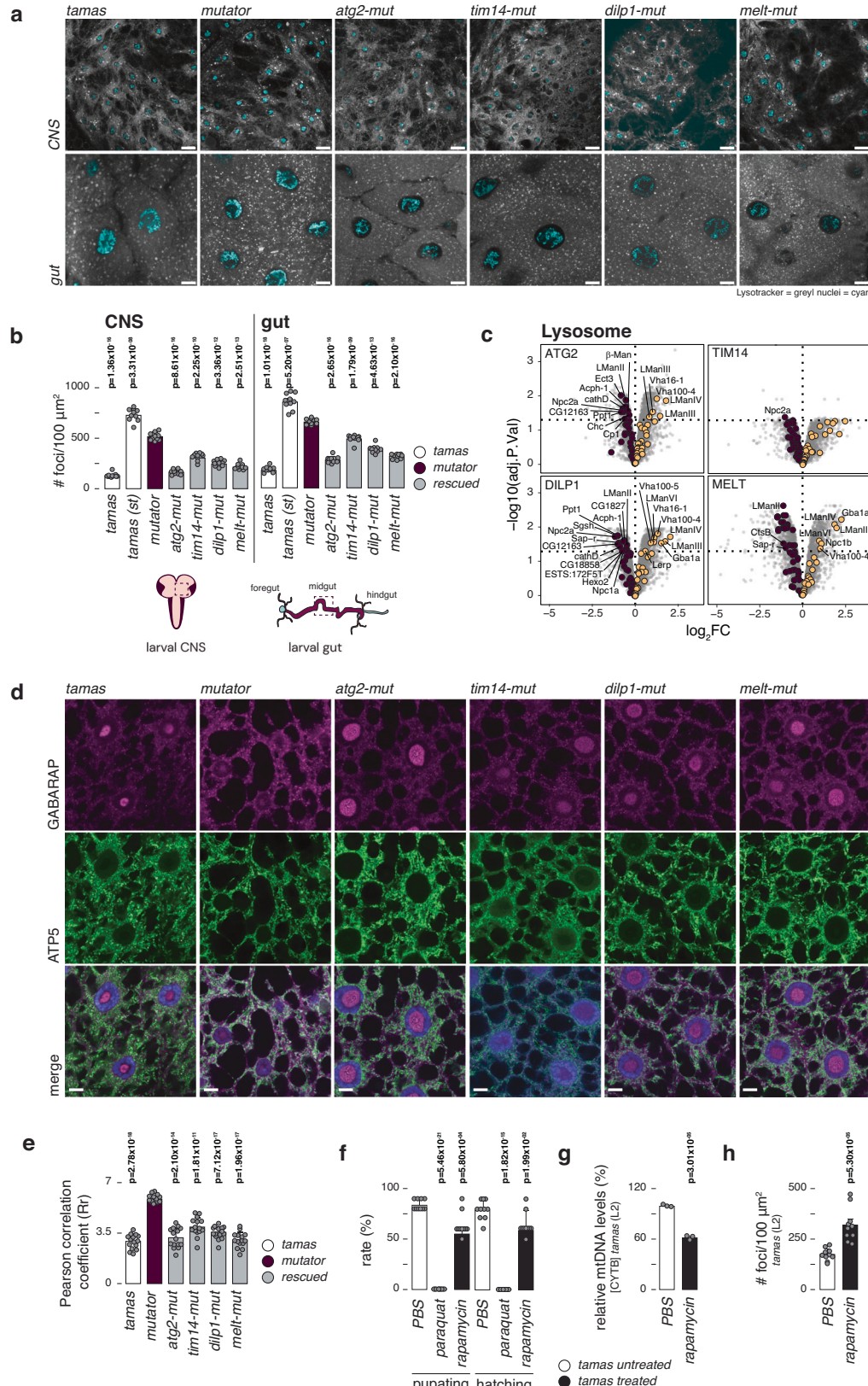

(Fig. 6c, d). Thus, although inhibition of autophagy at various stages rescued the mutator larvae, nucleation emerges as a key target for regulating mitochondrial turnover. Altogether, this suggests that mitochondrial dysfunction can induce excessive autophagy that drives the phenotype and that this response can be inhibited by targeting nucleation of the autophagy process.

## Discussion

Many diseases follow a common pattern, where an initial insult triggers cellular responses, such as altered nutrient sensing, activated detoxification mechanisms, inflammation, or cellular senescence. Often, these responses include metabolic adaptations, and while they initially may offer advantages, they alter the normal flow

**Fig. 5 | Increased macroautophagy in mtDNA mutator larvae. a** LysoTracker staining (grey) and Hoechst (cyan) in live brain tissue (CNS, top panels) and midgut cells (Bottom panels). Confocal images from brain and gut tissue from third instar fed larvae were taken directly after staining in lysosome-specific fluorescent and nucleus dye. Scale bar represents 10 μm. **b** Quantification of (a), showing the mean number of lysotracker foci per 100 μm² in the brain (top panels) and gut (Bottom panels) from control (*tamas*, white), *mutator* (plum), and rescued (grey) larvae. $N = 10$ independent biological replicates per genotype. **c** Volcano plots of respective genotypes against mutator larvae proteomes with the Lysosome category highlighted in plum (down) and yellow (up). Selected significant hits (FDR < 0.05) are annotated. Only rescued samples are shown. Control data are shown in the supplementary files. **d** Confocal images of fat tissue after Gabarap immunostaining, showing, autophagosomes in magenta, mitochondria in green (α-ATP-synthase) and nucleus in blue (DAPI). (Scale bar = 10 μm; Zoom = 1.5X). **e** Quantification of (d), using the red and green signal presented by Pearson correlation coefficient (Rr). *Tamas* (white) controls, *mutator* (plum), and rescued (grey) larvae are shown. Mean values of Rr are shown. $N = 15$ independent biological replicates were used per genotype. **f** Pupation and hatching rates of control (*tamas*) flies grown in the presence of paraquat or rapamycin (black). PBS was used as control (white). Mean values of hatched flies from number of embryos seeded are shown. $N = 10$ biologically independent samples with 10 embryos/sample. **g** Relative mtDNA levels in second instar *tamas* larvae after rapamycin (400 μM) treatment (black), using Cytb as mtDNA and rp49 as nuclear targets. Mean values relative to PBS are shown (white). $N = 3$ biologically independent samples with 3 technical replicates per genotype. **h** Number of lysosomal foci in *tamas* middle midgut cells after 400 μM rapamycin treatment (black) or untreated (white). Mean number of lysosomal foci (visualised by LysoTracker) are represented per 100 μm². Nuclei are stained by Hoechst staining. $N = 10$ independent biological replicates per genotype. Student's two-tailed T-test was performed in comparison to *mutator* (mut) samples for (b) and (e) and PBS-treated samples for (f), (g), and (h). P values < 0.05 are shown in bold. Error bars represent Standard deviation. Source data are provided in the main figure source data file.

of intracellular communication, and their sustained activation may ultimately prove detrimental to the cell. The pathological accumulation of mtDNA variants, as seen in mitochondrial diseases or during natural ageing, exemplifies this process, where the initial bioenergetic defect does not necessarily explain the severity or specificity of a rapidly deteriorating cellular function[11,15,47]. Using a rescue strategy in a *Drosophila* model with an accelerated accumulation of mtDNA mutations, we expose such a harmful response, where the excessive activation of autophagy contributes to the pathology. This overactivation can be prevented by normalising the expression of specific factors (e.g., *atg2*, *dilp1*) or reducing their expression, as in the case of *tim14* or *melted*.

Surprisingly, aside from members of the mitochondrial import machinery, none of the identified factors localised to mitochondria. This suggests that disease progression extends beyond a bioenergetic failure and likely involves a maladaptive signalling cascade[48,49]. In agreement, all but one rescued line retained a high mutational burden, suggesting that the mutation load alone is not the sole determinant of the mtDNA mutator phenotype.

Interestingly, our proteomic analysis revealed that the overall rescue mechanism was similar, with no rescued line presenting a majorly different proteomic profile. However, individual differences in the degree of modulation did exist. Accordingly, all identified factors are either part of autophagy or involved in its signalling via the IIS pathway or nutrient sensing. Additionally, *dilp1*, *inR*, and *glyS* have all been identified as longevity genes, slowing metabolism and modulating nutrient sensing[15,50]. Even *tim14*, the *Drosophila* homolog of mammalian stimulatory J-protein Pam18 (also known as DNAJC19), has recently been linked to mitochondrial clearance in response to misfolded proteins, albeit independent of nutrient sensing[51,52]. Tim14 is one of two membrane-bound cochaperones of the PAM complex that regulate ATP-hydrolysing activity of the mitochondrial heat shock protein mtHsp70[29]. It thereby regulates the activity of the Tim23 translocase. The deletion of Tim14 thus slows mitochondrial and cellular metabolism to reduce cellular stress. Additionally, *tim14*, together with prohibitin 2 and Taffazin, has been implemented in cardiolipin composition of mitochondrial membranes[53]. It is, therefore, possible that reducing mitochondrial-derived lipid supply to the growing phagophore is an effective way to reduce autophagy[54].

The function of *melted* or its mammalian homolog VEPH1 has yet to be established, although involvement in nutrient sensing via TSC1/2 and mTROC1 has been suggested, although the heterozygous deletion of *tsc2* could not rescue the mutator flies[33,35]. Interestingly, *melt-mut* animals were the only samples with a significant down-regulation of BNIP3, recently shown to be a critical TOR-dependent mitophagy receptor responsible for the programmed clearance of mitochondria in the fly germline[55]. It is thus possible that *melted* acts upstream of TOR early during fly development. This is supported by the observation that *melt-mut* larvae retained a much lower mutational burden than the mutator or the other rescue lines. Interestingly, though, while the number of unique mutations was less affected, meltmut larvae presented with less clonally expanded mutations, and although our sample size is limited, this suggests that *melted* affects the mitochondrial bottleneck. Thus, our results demonstrate that excessive autophagy can be prevented by targeting a single gene in multiple pathways, suggesting limited redundancy in the regulation of autophagy.

Our screen further exposed that the nucleation process of autophagy is an exclusive target for the rescue mechanism. In agreement, the nucleation-associated factor BCL2L13 rather than the autophagy initiation factor ULK1 was implicated with purifying selection in the mouse germline[56]. Notably, the pharmacological inhibition of nucleation with 3-MA in homozygous mtDNA mutator larvae also improved their phenotype, strongly supporting nucleation as a significant target for autophagy triggered by mitochondrial dysfunction.

What exactly triggers mitochondrial-induced autophagy is unclear, but our results suggest that a heightened mtDNA mutational burden and the associated OXPHOS defect create a sequence of excessive mitochondrial turnover and biogenesis. This vicious cycle will further increase the mtDNA mutation load or lead to clonal expansion of deleterious mtDNA mutations, as previously observed in patients with mitochondrial disease or individuals undergoing antiretroviral treatment[57,58]. Furthermore, excessive mitochondrial clearance has been implicated in the pathophysiology of autosomal dominant optic atrophy[59] and due to mutations in the mitophagy adaptor protein FBXL4[60]. Our finding that rapamycin, proposed to aid the clearance of damaged mitochondria, negatively affects both controls and mtDNA mutator flies supports the hypothesis that an elevated autophagic rate can lead to detrimental mitochondrial clearance and dysfunction. This potentially opens novel treatment strategies for patients with mitochondrial disease.

## Methods

### Drosophila stocks and culture
All experiments were performed with flies raised and kept at 25 °C and 60% humidity on a 12 h:12 h light: dark cycle on a standard yeast-sugar-agar medium. The isogenic deficiency kit fly strain and mutant lines were obtained from the Bloomington *Drosophila* Stock Center, Indiana, USA or Vienna *Drosophila* Resource Centre, Austria[27,61]. All fly strains used for this study are listed in Supplementary Data 1. We used *tamas* rescue and White Dahomey (*wDah*) flies as controls, ensuring an isogenic background. The mtDNA mutator and *tamas* control lines were described previously[21,62]. In brief, the endogenous *tamas* locus was replaced with a short attP site by 'ends-out' homologous recombination[63], followed by site-specific integration at the attP site of either wild-type (*tamas*) or D263A-containing (POLγ^exo-^) genomic

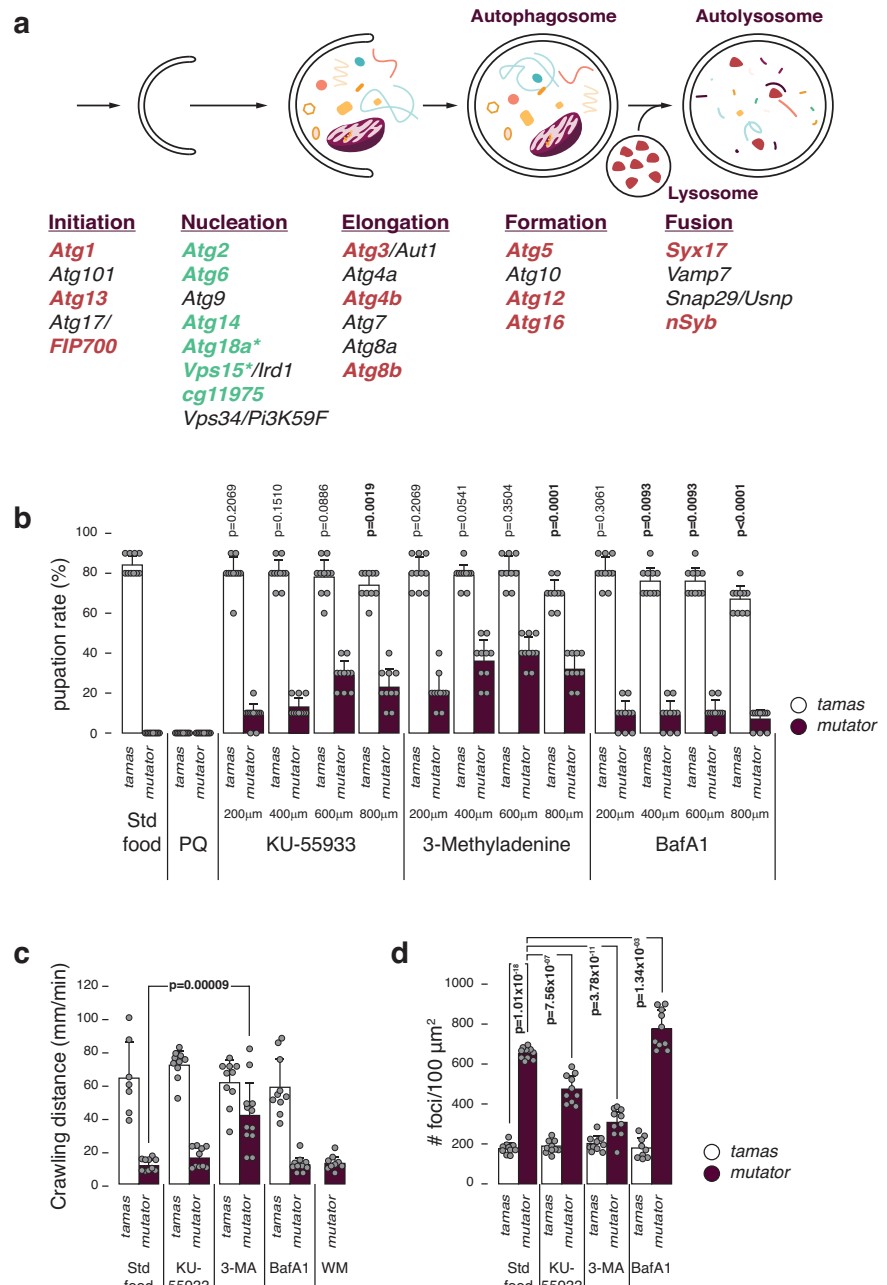

**Fig. 6 | Increased macroautophagy in mtDNA mutator larvae. a** Simplified scheme of the *Drosophila* autophagy pathway. Steps and involved genes are shown. Tested genes are coloured (red and green), with only green genes that rescued the mutator larva lethality. **b** Pupation rates after growing control and homozygous POLγ^exo- larvae in the presence of inhibitors specific to autophagy initiation (KU-55933), nucleation (3-methyladenine (3-MA)), or autolysosome formation (BafA1) at 3 different concentrations. PBS was used as control. Mean values of hatched flies from number of embryos seeded are shown. *N* = 10 biologically independent samples with 10 embryos/sample. **c** Crawling distance of third instar larvae grown in the presence of KU-55933, 3-MA and BafA1. Mean values of crawling distance in mm/min are shown. *N* = 10 biological replicates per genotype, except for *tamas* on standard food (*N* = 7), *mutator* on 3-MA (*N* = 12) and *mutator* on wortmannin (WM) (*N* = 9). **d** Quantification of lysosomal foci after LysoTracker and Hoechst staining in midgut cells from control and homozygous POLγ^exo- larvae grown in the presence of KU-55933, 3-MA and BafA1, showing the mean number of lysotracker foci per 100 μm². *N* = 10 independent biological replicates per genotype, except for *tamas* on BafA1 (*N* = 9). *Tamas* controls in white and *mutator* in plum. Student's two-tailed *T*-test was performed in comparison to *tamas* (standard food) for (b) and *mutator* (mut) (standard food) for (c) and (d). *P* values < 0.05 are shown in bold. Error bars represent Standard deviation. Source data are provided in the main figure source data file.

DNA. White Dahomey flies were a gift from Linda Partridge at the Max-Planck Institute for Biology of Ageing, Cologne, Germany.

### Deficiency Screening
To set up a screen cross, ten virgin females were crossed with five males and reared at 25 °C. After three days, parental flies were discarded. Ten three-day-old virgin females and five male progenies of the correct genotype were used for the appropriate crosses (Supplementary Fig. 1). Mutator females for crossing were selected from fresh *wDah* backcrosses. The entire screen was performed twice, using *glossy* or *curly-gfp* balancers with the same outcome.

## Locomotion performance test

A negative geotaxis assay was used to test fly locomotor performance. Ten flies were placed in a plastic column and gently tapped to the bottom. After 30 s, the flies at the top of the column (Ntop) and those remaining at the bottom (Nbot) were counted. Three trials were performed at 30 s intervals. The performance index was defined as $(1/2*(10 + Ntop\text{-}Nbot)/10)$.

## Larvae crawling test

Third-instar larvae were placed on a 15 cm Petri dish filled uniformly with agar-water media. A simple camera system on a stand was used to record low-magnification time-lapse images of the crawling larvae at one frame per second for 1 min. The trajectory was measured using Fiji software[64] to calculate the larvae speed.

## DNA extraction

Third instar larvae were collected in Lysing Matrix D tubes (MP Biomedicals, 116913100) containing Buffer A (100 mM Tris-HCl, 100 mM EDTA, 100 mM NaCl and 0.5% SDS) and homogenised in a FastPrep Tissue Homogenizer (MP Biomedicals). Lysates were collected and incubated at 65 °C for 30 min, followed by incubating with Buffer B (2:5 of 5 M potassium acetate and 6 M lithium chloride) on ice. After centrifugation at 13,500 x $g$ for 15 min at room temperature, supernatants were collected and mixed with 0.6 volumes of isopropanol and centrifuged at 21000 x $g$ for 10 min. Supernatants were removed, and the pellets were washed with 800 µl ethanol (70% v/v). After centrifugation, supernatants were carefully discarded, and pellets were dissolved in DNase/RNase-Free Water.

## mtDNA Quantification

qPCR was performed with 5 ng of DNA per well according to the following recipe: 1 µl DNA, 5 µl Master Mix, 1 µl 10 mM forward and reverse primer mix, and 3 µl water. qRT-PCR was performed on a QuantStudio 6 (Thermo Fisher Scientific), with Platinum SYBR Green qPCR supermix-UDG (Thermo Fisher Scientific). Normalised data was used to determine the relative levels of mtDNA molecules according to cycling threshold analysis (ΔCt). Primers are listed in Supplementary Data 5.

## Random capture mutation assay (RMC)

RMC was performed essentially as described previously[40,65]. Total DNA from ten third instar larvae was digested using TaqI restriction enzyme (New England Biolabs, R0149S) for 8 h, at 65 °C, followed by qPCR across TaqI sites in either mt-tRNA-Arg or mt-CytB (Supplementary Data 5). A Platinum SYBR Green qPCR supermix-UDG (Thermo Fisher Scientific) was used on a QuantStudio 6 (Thermo Fisher Scientific) according to the following program: 10° at 95 °C, 45 x[30 at 95 °C, 60” at 60 °C, 15” at 72 °C] +60” at 72 °C. Relative mutation loads were calculated by normalisation to total mtDNA levels using primers within mt-COXIII. Normalised data was used to determine the relative levels of mutated mtDNA molecules according to cycling threshold analysis (ΔCt).

## Cloning and sequencing for mutation frequency determination

DNA from 3rd instar larvae from two (three for *melt-mut*) independent crosses were used to amplify a 825 nt fragment of mtDNA, using a high-fidelity Phusion Polymerase (Thermo Fisher Scientific), following the manufacturer's instructions with the following program: 30 s at 98 °C, 30x [5 s at 98 °C, 15 s 58 °C, 2 min 70 °C], 5 min 70 °C. Primer sequences are listed in Supplementary Data 4. PCR products were purified using AmPure XP (Beckman Coulter Life Sciences), before cloning into the Zero Blunt TOPO cloning vector following manufacturer's instructions (Thermo Fisher Scientific). An appropriate number of colonies was picked from overnight LB culture plates and used to inoculate 2 ml of an LB-kanamycin culture. Plasmid preparations were performed as follows. The overnight cultures were pelleted and resuspended in ice-cold alkaline lysis solution I (50 mM glucose, 25 mM Tris-Cl pH 8.0, 10 mM EDTA pH8.0), followed by lysis in freshly prepared alkaline solution II (0.2 N NaOH, 1% (w/v) SDS), and neutralisation in ice-cold alkaline solution III (3 M potassium acetate pH5.2). Lysates were centrifuged for 5 min at 15000 x $g$, 4 °C, followed by EtOH precipitation. 0.5 µl of each preparation was used for M13 sequencing on an ABI Sanger sequencer, using Bigdye 3.1 technology (Thermo Fisher Scientific).

Resulting sequences were aligned to the *wDah* mtDNA sequence (KP843845.1) using Geneious Prime 2024.05 software (http://www.geneious.com/) and inspected individually for variants. All sequenced clones contained the 4853 G > A polymorphism, distinguishing *wDah* (KP843845.1) from *w1118* (NC_024511). Sequences were excluded when multiple base calls were made per position.

## RNA isolation, reverse transcription, and quantitative PCR

mRNA was extracted from ten third-instar larvae using TRIzol (Thermo Fisher Scientific) and quantified with a Qubit fluorometer (Thermo Fisher Scientific). Reverse transcription for qRT-PCR analysis was performed using High-Capacity cDNA Reverse Transcription Kit (Thermo Fisher Scientific), according to the following program: 7 min at 95 °C, 41x [15 s at 95 °C, 1 min at 60 °C]. qRT-PCR was performed with Platinum SYBR Green qPCR supermix-UDG (Thermo Fisher Scientific) and gene-specific primers, using QuantStudio 6 (Thermo Fisher Scientific) (Supplementary Data 5).

## Mitochondrial preparations

For enrichment of larvae mitochondria, ten larvae per genotype (3 independent samples per genotype) were homogenised with a Teflon-coated Dounce homogeniser (10 strokes at 700 rpm) on ice in STE-enrichment buffer [250 mM sucrose, 5 mM tris, and 2 mM EGTA (pH 7.4)] with 5% bovine serum albumin (BSA). Detritus and large cellular components were removed by centrifugation at 1000 x $g$ for 10 min at 4 °C, and mitochondria were enriched through differential centrifugation twice at 3000 x $g$ for 10 min and in STE buffer without BSA at 7000 x $g$ for 10 min.

## Isolated respiratory chain complex activities

The respiratory chain enzyme activities of complex I (NADH: CoQ reductase), complexes I and III (NADH: cytochrome c reductase), complex II (succinatdehydrogenase) complexes II and III (succinate: cytochrome c reductase), complex IV (Cytochrome c oxidase), and citrate synthase (CS) were determined as described with some modifications[66]. All enzyme activities were determined at 35 °C on an Indiko automated photometer (Thermo Fisher Scientific) and are expressed as units per unit of CS activity in the mitochondrial suspension. For complexes CI and I + III mitochondria were resuspended in 50 mM $KH_2PO_4$, 5 mM $MgCl_2$, 5 g/L HSA, 0.2 mM KCN to a final pH of 7.5.

For CI activity 1.2 mg/L antimycin A and 0.12 mM coenzyme $Q_1$, and 0.15 mM NADH were added before measuring the decrease in absorbance at 340 nm for 1 min before and after the addition of 2 mg/L rotenone.

For CI + III activity 0.12 mM cytochrome $c$ (oxidised form) and 0.15 mM NADH were added before measuring the increase in absorbance at 550 nm for 1 min before and after the addition of 2 mg/L rotenone.

CII activity was measured in 50 mM $KH_2PO_4$, 12.5 mM $MgCl_2$, 62.5 mM succinate, 0.5 mM KCN, 0.75 mM 2,6-Dichloroindophenol sodium salt hydrate, 30 mg/L antimycin A, 2 mM coenzyme $Q_1$, and recording the decrease absorbance at 600 nm.

CII + III activity was measured using rotenone-treated (see CI + III) mitochondria in 50 mM $KH_2PO_4$, 30 mM succinate, 7.5 mM $MgCl_2$ (pH 7.2) and monitoring the absorption at 550 nm for 2 min.

CIV activity was measured in mitochondria pretreated with 1 g/L digitonin and 50 mM $KH_2PO_4$ (pH 7.5). Background was measured in

the presence of 2 mg/L rotenone and 30 mg/L cytochrome *c* (reduced form) (pH 7.5). Absorbance was measured at 550 nm for 1 min.

Reverse CV activity (ATP hydrolysis) was measured using a spectrophotometric method previously described[67]. Briefly, isolated mitochondria were resuspended and incubated for 3 minutes at room temperature in pre-assay buffer (250 mM sucrose, 10 mM KCl, 5 mM $Mg_2Cl$, 1 mM EGTA, 1 g/L BSA, 50 mM Tris (pH 8.25), 1 μM antimycin A, 1 mM phosphoenolpyruvate, 0,01% (w/v) DDM and 3 μM AP5A). After, 10 μL of pretreated mitochondria were added to 180 μL of assay medium (pre-assay medium with 0,4 mM NADH, 10 U/mL lactate dehydrogenase and 25 U/mL pyruvate kinase) and incubated for 5 min at 37 °C. Following incubation, ATP at a final concentration of 1 mM was added to the mixture and decrease in absorbance at 340 nm was measured for 10 min. Finally, 5 μM Oligomycin was added, and activity was measured for 10 min. Oligomycin sensitive activity was calculated and normalised to citrate synthase activity.

### TMRE staining
Whole gut from 3rd instar larvae was dissected in room temperature Schneider's *Drosophila* medium (Gibco, 21720-020) and stained for 20 min with 1 μM Tetramethylrhodamine, ethyl ester (TMRE) (Thermo Fisher Scientific, T669), followed by two five-minute washes in Schneider's *Drosophila* medium and imaging on a Zeiss LSM confocal microscope and analysed using Image J software (Fiji). Calculations were performed on 10 images per gut.

### Aconitase activity measurements
Mitochondrial aconitase activity was measured in larvae using an Aconitase Activity Assay kit (Abcam, ab109712). 40 μg of the mitochondrial suspension was added to 50 μl of diluted substrate solution. 200 μl of assay mixture was added, and the absorbance at 340 nm was monitored once per minute. Aconitase activity was calculated according to the manufacturer's protocol. Mitochondrial protein content was determined using a BCA assay kit (Thermo Fisher Scientific). All measurements were performed as at least two technical and 3 biological replicase.

### Transmission electron microscopy
Muscle tissue from third instar larval was fixed in 2.5% glutaraldehyde and 1% paraformaldehyde in phosphate buffer and stored at 4 °C. Following the primary fixation, the samples were rinsed with phosphate buffer, followed by post-fixation in 2% osmium tetroxide in phosphate buffer at 4 °C for 2 h. The samples were then subjected to stepwise ethanol dehydration followed by stepwise acetone/LX-112 infiltration and embedded in LX-112. Ultrathin sections (approximately 60 to 80 nm) were prepared using an EM UC7 ultramicrotome (Leica Microsystems). The ultrathin sections were contrasted with uranyl acetate followed by Reynolds lead citrate and examined in a Tecnai Spirit BioTWIN TEM (Thermo Fisher Scientific) operated at 100 kV. Digital images were acquired using a 2kx2k Veleta CCD camera (EMSIS GmbH).

### Immunostaining and confocal microscopy
For ATP synthase and GABARAP immunolabelling, brains and fat tissue from third instar larvae were dissected in PBS, fixed for 20 min in 4% formaldehyde, washed for 5 min, and saturated for 1 h in 0.5% BSA (Merck), 0.1% Triton (Sigma-Aldrich, P1379) in PBS (PBTB). Primary anti-ATP synthase (Abcam, Ab110275 at 1:300) and GABARAP (Cell Signaling Technology, #13733 at 1:300) were diluted in PBTB and incubated over-night at 4 °C and later washed for 1 hour in 0.1% Tween20 (Sigma, P1754) in PBS (PBTw). Secondary antibody Alexa Fluor 488 (Invitrogen, A-32723) and Alexa Fluor 568 (Invitrogen, A-11011) were incubated for 2 h, followed by a 1-hour wash in PBTw. Preparations were mounted in Vectashield/DAPI (Vector, H-1200). An LSM880 Zeiss confocal microscope was used for imaging. For the

GBARAP experiment, the third instar larvae were fed with Bafilomycin A1[68], 800 nm (Cell Signalling Technology, #54645) for 24 h. Antibody information can be found in Supplementary Data 6.

### Lysotracker staining
Brain and gut tissue from fed or starved (6 h starvation in agar-water medium) third instar larvae were dissected in PBS and incubated for 2 min in 100 μM LysoTracker Deep Red (Invitrogen, L12492), 1 μM Hoechst 33342 (Thermo Scientific,62249) in PBS. Samples were transferred to PBS on glass slides, covered, and photographed live on a Zeiss Confocal LSM880 (Lysotracker excitation wavelength 647/668).

### Cathepsin B staining
Whole gut from 3rd instar larvae was dissected in room temperature PBS and incubated for 10 min in 100 μM Magic Red (Immunochemistry Technologies, #928), 1 mM Hoechst 33342 (Thermo Fisher Scientific, #62249) in PBS, followed by three five-minute washes in PBS and visualised on a Zeiss LSM880 confocal microscope (wavelength 603/622) and analysed using Image J software (Fiji). Calculations were performed on 10 images per gut.

### Western blotting
Protein extractions from ten third instar larvae were performed in 15 μl RIPA buffer (150 mM NaCl, 0.1% SDS, 50 mM Tris, 0.5% Sodium deoxycholate and 1% Triton X-100) per mg of sample. Samples were homogenised and incubated on ice for 30 min, followed by a 10 min centrifugation at 13,500 x *g*. Supernatants were collected and denatured with NuPAGE™ LDS Sample Buffer (4x) (Invitrogen, NP0008) for 10 min at 70 °C. For Atg8a lipidation, ten third instar larvae were incubated at 100 °C for 3 min in 15 μl PBS (Sigma-Aldrich, P4417) and 5ul NuPAGE™ LDS Sample Buffer (4x) per mg of larvae. Samples were then homogenised, followed by a second incubation at 100 °C for 5 min. Supernatants were collected by centrifugation for 5 min at 21,300xg.

Proteins were loaded onto NuPAGE™ 4–12% Bis-Tris Mini Protein Gel (Invitrogen, NP0321), transferred with the iBlot™ PVDF Transfer Stacks (Invitrogen, IB401001), following blocking with 5% non-fat milk (Semper) and decorated with appropriate antibodies. Blots were visualised with Clarity™ Western ECL Substrate (Bio-Rad, 1705061). Antibodies used: anti-NDUFS3 (AbCam, ab14711 at 1:2000), anti-β-actin (Cell Signaling, #4967 at 1:2000), anti-GABARAP (Cell Signaling Technology, #13733 at 1: 2000).

As secondary antibodies either Mouse IgG-HRP (Cytiva, NA9310V at 1:5000) or Rabbit IgG-HRP (Cytiva NA9340V at 1:5000) were used. For Bafilomycin A1 treatment, 400 μM of Bafilomycin A1 (Cell Signaling Technology, #54645) in PBS or DMSO was added to the third instar larvae food vial. Samples were collected for Western blot analysis after 24 h of treatment.

### Proteomics sample preparation for rescue larvae
Sample preparation was performed as previously described[69,70]. Frozen third instar larvae were collected in Lysing Matrix D tubes (MP Biomedicals, 116913100) containing 1% SDC (sodium deoxycholate) and 100 mM Tris pH 8.5 and homogenised for 20 s in a FastPrep Tissue Homogeniser (MP Biomedicals). Supernatants were collected and boiled for 10 min at 96 °C. 80 μl of the solution were then sonicated in a Covaris ultrasonicator. Protein concentration in the supernatant was determined using the BCA (bicinchoninic acid) assay. For protein reduction and alkylation, chloroacetamide (CAA) and Tris(2-carboxyethyl) phosphine (TCEP) were added to a final concentration of 40 mM and 10 mM, respectively. Proteins were then incubated at 45 °C for 5 min. After adding Trypsin (1:100 w/w, Sigma-Aldrich) and LysC (1/100 w/w, Wako), proteins were subjected to overnight digestion at 37 °C. 200 μl of 1% TFA in isopropanol was added to the samples to quench protein digestion. Subsequently, peptides were loaded onto

SDB-RPS StageTips (Empore), followed by washes with 200 µl of 1% TFA in isopropanol and 200 µl of 0.2%TFA in 2%ACN. Peptides were eluted with 60 µl of 1.25% NH$_4$OH in 80% ACN and dried using a SpeedVac centrifuge (Eppendorf). Dried peptides were resuspended in A* (0.2% TFA in 2%ACN). Peptide concentration was estimated using a NanoDrop, and 250 ng of peptide material was used for individual measurements.

## Proteomics sample preparation for larvae without mutator allele

*Drosophila* samples were transferred from the −80 °C freezer into lysing matrix tubes containing 400 µl Preomix buffer and homogenized using a bead homogenizer for 20 s. The samples were then centrifuged for 10 min at 10,000 x *g*, and the supernatant was collected. Samples were boiled for 10 min at 96 °C with a shaking speed of 300 rpm. Subsequently, 80 µl was processed using an Ultrasonicator (Covaris) with the following settings: 300 s, 450 Watts PIP, 50% DF, 200 CBP, and 225 AIP. Protein concentration was determined using a BCA assay kit, and the mixture was incubated at 45 °C for 10 min. For protein digestion, 50 µg from each sample was digested with LysC/Trypsin at a 1:100 ratio and incubated at 37 °C for 18 h with mild shaking, then quenched with 200 µl of 1% TFA in isopropanol and briefly vortexed. Peptides were cleaned up using SDB-RPS stage tips, washed with 200 µl of 1% TFA in isopropanol, followed by 200 µl of 0.2% TFA in 2% ACN. Peptides were eluted with 70 µl of 1.25% NH$_4$OH in 80% ACN, dried in a speedvac for 50 min at 45 °C, and resuspended in 15 µl of buffer A* (0.2% TFA, 2% ACN). Peptides were further resuspended using a thermo shaker for 10 min at 1750 rpm, briefly spun down, and their concentration measured using a Nanodrop. Peptides were diluted with buffer A* to a final concentration of 100 ng/µl for individual measurements. Of these, 250 µg were loaded onto Evotips as described above. Peptides were then separated by liquid chromatography using an Evosep One system with an 60 samples per day (SPD) method (21 min gradient) at factory settings. Peptides were eluted into a timsTOF Pro 2 mass spectrometer (Bruker) operated as described previously[71]. In short, twelve py_diAID-optimised dia-PASEF scan windows were deployed at a cycle time of 1.4 s. The settings included a mass-to-charge (m/z) range of 350–1200 and an ion mobility (IM) range of 0.70–1.30. Collision energy was decreased as a function of the IM from 59 eV at $1/K_0 = 1.6$ V cm$^{-2}$ to 20 eV at $1/K_0 = 0.6$ V cm$^{-2}$. Calibration of the IM dimension was performed with three Agilent ESI Tuning Mix ions (m/z, 1/K$_0$: 622.02, 0.98 V cm$^{-2}$, 922.01, 1.19 V cm$^{-2}$, 1221.99, and 1.38 V cm$^{-2}$). Data was searched with DIA-NN v1.8 against the same FASTA file as described above[72].

## LC-MS/MS and proteomics data analysis

For liquid chromatography (LC), an 88 min Evosep gradient was used and directly coupled online with the Exploris 480 mass spectrometer (Thermo Fisher Scientific). Data acquisition was carried out in DIA mode with a scan range of 300–1300 m/z and a resolution of 120,000. The AGC was set to 3e6, and the maximum injection time was 60 ms. For precursor fragmentation, HCD was used with NCD set at 25.5%, 27.5%, and 30%. Fragment ions were analysed in 40 DIA windows at a resolution of 30,000, while the AGC was maintained at 1e6. DIA raw files were processed using Spectronaut (v16.3) using default settings. Protein intensities were further analysed in R version 4.3.1, using limma linear modelling differential expression analysis for statistical testing.

## Treatment of Drosophila larvae with autophagy modulating drugs

Hatching rates were determined by mating 3-day-old virgin males and females with appropriate genotypes for 1 h at 25 °C on freshly prepared apple juice agar plates. Resulting eggs were counted and transferred to appropriate vials containing standard food. Pupation and hatching rates were determined 4–7 days later.

For drug treatments, seven-day-old virgin flies (ten females and 5 males per vial) were mated on standard food, supplemented with 400 µM paraquat (Sigma-Alrdich, #75365-73-0), rapamycin (Sigma-Alrdich, #R0395, KU-55933 (Selleckchem,#S1092), 3-methyladenine (Selleckchem, #S2767) or Bafilomycin A1 Cell Signaling Technology, #54645). All drugs were dissolved in PBS and added twice per day to the food.

## Statistics and reproducibility

A two-tailed *T*-test was used for statistical testing throughout the manuscript unless stated otherwise. An ANOVA test was used for gene expression analysis of proteomic data. Significance of functional categories was determined by the Fisher's exact test. No statistical method was used to predetermine sample size. No data were excluded from the analyses. The experiments were not randomised. The Investigators were not blinded to allocation during experiments and outcome assessment.

## Reporting summary

Further information on research design is available in the Nature Portfolio Reporting Summary linked to this article.

## Data availability

The mass spectrometry proteomics data generated in this study have been deposited to the ProteomeXchange Consortium via the PRIDE[73] partner repository with the dataset identifier PXD041418. Source data are provided with this paper.

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

## Acknowledgements
This study was supported by grants from the Swedish Research Council (VR2022-01287 (AWr) VR2023-07091(AWr), VR2019-01154(AWe)), the Novo Nordisk Foundation (NN0082202(AWr)), the Knut and Alice Wallenberg Foundation (KAW2019.0109 (AWr), KAW2020.0228(AWe)), the Region Stockholm (RS2022-0708 (A.Wr.)), and a Karolinska Institutet consolidator grant (2-190/2022 (A.Wr.)), the Swedish state under the ALF-agreement (FoUI-955096 (A.We.)). MM is supported by the Max-Planck Society for Advancement of Science. F.A.R. is an EMBO post-doctoral fellow (ALTF 399-2021 (FAR)). We thank Dirk Wischnewski from the Max-Planck Institute of Biochemistry for proteomic sample preparations. We thank Linda Partridge and Sebastian Grönke from the Max-Planck Institute for Biology of Ageing for donating the *WDah* strain. Fly stocks were obtained from the Bloomington Drosophila Stock Centre (NIH P40OD018537) and Vienna Drosophila Resource Centre.

## Author contributions
Funding: Anna Wredenberg (A.Wr.), Anna Wedell (A.We.), M.M., P.F.C. Concept and Design: C.F., N.E.F., A.Wr., Experiments: NEF, FAR, KC, Alissa Wilhalm (A.Wi.), F.M.H., Z.G., T.B-O., D.A., Data evaluation: N.E.F., FAR, T.B-O., A.We., M.M., P.F.C., C.F., A.Wr., Writing: N.E.F., C.F., A.Wr.

## Funding

## Competing interests
The authors declare no competing interests.
