## [Transparent Peer Review file · Nature Communications]

Preventing excessive autophagy protects from the pathology of mtDNA mutations in *Drosophila melanogaster*

Corresponding Author: Professor Anna Wredenberg

Version 0:

Reviewer comments:

Reviewer #1

(Remarks to the Author)

In the manuscript titled " Preventing excessive autophagy protects from the pathology of mtDNA mutations " Najla El Fissi et.al demonstrate that some critical pathways including nutrient sensing, insulin signalling, mitochondrial protein import, and autophagy that can rescue from the lethal phenotype induced by mtDNA mutations. The findings presented in this manuscript are potentially interesting to the readership, however, some data do not sufficiently support the conclusion.

Major comments

1. In Figure 4A, an increase in lysosomes (as indicated by lysotracker staining) was observed in mtDNA mutator larvae. To fully assess the integrity and activity of lysosomes, additional methods should be employed in both the mtDNA mutator larvae and the rescue lines.
2. Protein levels may not precisely reflect mitochondrial activity; hence, it is advisable to measure ATP production, mitochondrial ROS, and mitochondrial membrane potential in the mtDNA mutator larvae and rescue lines to obtain a more accurate assessment of mitochondrial function.
3. In Extended Data Figure 9C, the ratio of Atg8a-I/Atg8a-II should be measured to assess the activity of autophagy.
4. The mitochondrial morphology could be analyzed by confocal microscopy or transmission electron microscopy in the mtDNA mutator larvae and rescue lines.
5. It is interesting yet puzzling that only the nucleation, but not other processes of autophagy rescue the mutator lethality, it should be explained and discussed in detailed.

Reviewer #2

(Remarks to the Author)

In this paper, Fissi et al. utilized a *Drosophila* model to identify genes that help counteract the developmental lethality caused by an mtDNA polymerase mutant D263A that is defective in proofreading. By conducting a deletion screen, they found removing one copy of several genes associated with nutrient sensing, insulin signaling, mitochondrial protein import, and autophagy that rescued the lethality of homozygous D263A. They then performed proteomics to profile the difference between D263A with and without the hemizygote mutation of the rescue genes, tested a few other autophagy components, and concluded that the reducing autophagy, particularly the components involved in the nucleation step, attenuated the impact of accumulated mtDNA mutations in the D263A mutant.

This work harnesses the power of *Drosophila* as a model system for genetic screening. Various techniques including proteomics and imaging were adapted to support their findings. However, the key question "how and why reducing autophagy nucleation alleviates the developmental lethality linked to D263A mutants and its relation to mtDNA mutations" remains unaddressed. Many data lacked proper explanations or speculations, which made the manuscript challenging to read and comprehend. Furthermore, the lack of key information in the figure legends and methods section, such as the specific genotypes and samples used for each experiment, makes it difficult to interpret the results.

While the deletion mapping is neat and the omics data could be valuable for other researchers, some of the major conclusions need further validation and certain observations would benefit from additional investigations. One would expect to gain more understanding of the topic from a manuscript in Nature Communication. Below I list some of my concerns.

Major points:

1. The whole manuscript is based on the developmental lethality phenotype of the homozygote D263A mutant used for this study. However, whether the lethality results from the accumulation of mtDNA mutation was not addressed. This is especially important because there are other fly mutator models showing this is not the case. For example, Samstag et al (PLoS Genetics, 2018) showed that D263A/polG deficiency flies accumulated mtDNA mutations, yet >60% of them survived to adulthood. Another fly mutator model mito-APOBEC (Andreazza et al, Nature Communication, 2018) also showed ubiquitous expression of mito-APOBEC from early development is highly mutagenic for mtDNA, but flies reach adulthood although they have reduced lifespan. Interestingly, although APOBEC1 flies carry a high level of mtDNA mutations, they don't show a reduced mtDNA copy number. Additionally, it is known that Drosophila can tolerate high levels of mtDNA with large deletion (e.g. Lingenhohl et al, PNAS 1992). These made me wonder whether the developmental lethality observed with the D263A homozygous mutant used for the current manuscript is caused by the polGEXO- itself, a higher level of mtDNA mutations carried by this D263A mutant compared to other mutator models, or due to other factors in the genetic background. Hence, it is important to test the effect of four key genes (atg2, tim14, melt and dilp1) in a different mutator model to show whether the rescue is true in flies carrying high levels of mtDNA mutations, independent of how the mutations were introduced. The authors could for example express mito-APOBEC1 and examine whether removing one copy of atg2, tim14, melt or dilp1 rescued the defects associated with mito-APOBEC1 expression. This experiment will strengthen the manuscript's focus on the pathology of mtDNA mutations rather than on POLG1, as indicated by the title.

2. The manuscript showed an interesting observation that only the nucleation step of autophagy is crucial for the rescue. However, it falls short of addressing why upstream or downstream of the same pathway does not elicit the same effect. It is a selling point for the manuscript, and one would expect authors to provide more insights/explanations for such discrepancies. For example, it raises questions about whether the nucleation step triggers additional processes besides autophagy, or if the nucleation components are rate-limiting factors of the entire autophagy process, so removing one copy of these factors are more likely to produce a phenotype than components in other autophagy steps. All experiments conducted involved deletion or RNAi. It would be interesting to do gain-of-function experiments to test whether upregulating autophagy is sufficient to reduce mitochondrial DNA copy number, and mitochondrial mass and cause the lethality from the pupa to adult transition in the absence of the mutator allele.

3. It is not clear whether deficiency + D263A or mutant allele of the genes of interest + D263A were used for all phenotyping, proteomic analysis and imaging presented in Fig 2-4. If deficiency lines were used, the difference reported could stem from other genes deleted in the same def lines used. If p-element mutants were used, the authors need to explicitly state the lines used in the main text and the corresponding figure legends. Additionally, the validation of the mutants used as true KO mutants is lacking (e.g. 17156 has the p-element insertion in the 5' UTR of Atg2 and does not seem to reduce the mRNA level of Atg2 as shown in Fig S2C). Furthermore, it is not shown whether the proteomic and lysosome changes observed with hemizygotes (Fig 2 and 4) are linked to mtDNA mutation levels. In other words, the authors should present the mass spec proteomic data of hemizygotes in the absence of the D263A mutation, at least for some genes.

4. Did four rescue lines rescue the D263A in the same manner? The authors have reported their differences in various assays (mtDNA mutation levels, PGC-1 α expression etc), but surprisingly, they all seem to down-regulate autophagy/lysosome. I found this puzzling, and the authors do not provide proper explanations or hypotheses.

Minor points:

1. The mtDNA mutation level in Fig 1I should be presented in actual values, not in relatives (refer to the major point 1 to see why such information is needed). This also applies to Fig 1H and J.
2. It would be nice to state what assigned color labels on the confocal images directly to help read Fig 3B, 4A and 4D.
3. The images presented in Fig 3B and 4A missing key information: which part of gut/CNS and what cell types are in the representative image? This is essential information as cells in different parts of the gut/brain are very different. Are all the images in those panels from the same cell type and tissue part? If yes, please show markers to confirm this. An alternative approach is to make genetic clones to show the differences in neighboring cells.
4. What is the difference between wDah and tamas, and which one should be considered as the control? Why were wDah data only shown in Fig2 and 3A, but not Fig 1 and 4? This is important since 3A showed wDah had similar mito protein content as mutator but tamas control did not.
5. The method section should contain details describing how different complex activities were measured (Fig 1F).
6. Fig 2D and 3E, please label the significantly up/down regulated protein as 4C and mark PGC1 α in 3E since they mentioned it in the main text.
7. Fig 3A what is the $-\log_{10}$ and ratio rel?
8. Fig 4D, why fat tissue? Why were CNS and gut cells chosen for Fig 4A and 3B?
9. The nature of D263A/tamas control and how they were generated should be provided in the main text or the method section. Citing the Bratic et al (2015) paper, which contains a detailed description of how the D263A mutant was generated, is not sufficient. A brief description is needed in this manuscript, so the readers don't need to refer to other publications to understand the nature of the mutant.
10. Fig S4A, B, are they mtDNA copy numbers or mutation loads? The figure legend does not seem to match.
11. There is no data to support the statement on the melted rescue (line 113) being "early in the fly development". In fact, according to Flybase, the expression of melted continued to be low and only upregulated to moderate expression during the pupae stage. The author should provide a proper explanation or data to support this statement.
12. The melt rescue flies have much-reduced mtDNA mutation levels compared to other rescuers and D263A alone. Does this line live longer than the other rescuers after eclosion?

(Remarks to the Author)

In their manuscript entitled "Preventing excessive autophagy protects from the pathology of mtDNA mutations", Fissi et al. study the consequences of mitochondrial DNA mutations resulting from mutant mtDNA polymerase in drosophila. The authors carried out a genetic screen in isogenic hemizygous deficiency strains to identify genes that can rescue the mtDNA mutator fly phenotype. They identified nine genes, part of autophagosome formation, mitochondrial protein import, insulin-like growth signaling, and nutrient sensing. To understand the courses underlying rescue, the authors analyzed by mass spectrometry the proteomes of different fly tissues with and without rescue. Lastly, the authors monitor specific pathways to observe reduced mitochondrial mass and increase macroautophagy in mtDNA mutator larvae. The manuscript is well written and the data are of high quality. The manuscript builds on established technologies to identify rescue genes and describe resulting proteome changes. However, additional validation or mechanistic study of observations is necessary to extract additional new insight.

Main comments

- 1) The overall role of autophagy in protecting from mtDNA mutations remains unclear. Ext Data Fig 9 C, D: It is unclear what was quantified, are the shown values ratios of the lipidated versus non-lipidated Atg8 and or some comparison between DMSO versus BafA? The overall experiment is difficult to interpret and a more clear description of the method and interpretation would be helpful. This includes the vastly different effects of the tamas controls across in the two Western blots. In addition, how can the observed Atg8 degradation in tamas upon BafA treatment be explained? This appears counterintuitive. Overall, the data does not appear to be strong enough to support the main claim of the title that preventing excessive autophagy is protective for mtDNA mutations. Additional experiments are required to show the autophagy effects and particularly to support the point of "excessive" autophagy.
 - 2) The lack of identifying Parkin in the screen is not sufficient to rule out a role for mitophagy in the observed reduction in mitochondrial mass. Additional experiments that directly target mitophagy (e.g. via Pink1, Parkin) are required to make this point.
 - 3) The identified rescue genes are a key outcome of the manuscript. Strikingly, most genes hit the same pathways, strongly supporting these as correct. Considering the important impact of the remaining genes of the import and nutrient sensing pathway to the manuscript, additional validation experiments depleting other components of these pathways would be important.
 - 4) The proteomics data is an important asset of the manuscript. Please make the data more accessible by providing full processed data as supplementary table.
- Minor comments:
- 5) It would be helpful to the reader to have an explanation for why the 3rd chromosome specifically was screened.
 - 6) Are the effects observed in Fig 3A,D statistically significant?
 - 7) Fig 1F: y-axis description is missing
 - 8) Fig 1D,H: points should be used rather than commas in numbers (e.g. "1.5" rather than "1,5"). This also occurs in other figures.
 - 9) Fig. 2D, Fig. 3E: it would be helpful to indicate the base of the logFC shown (i.e. log₂FC).
 - 10) Fig. 3A: There is a misplaced textbox "-log₁₀(adj.P.Val)" at the bottom of the panel.
 - 11) Many figures only have very selected statistical information indicated and p-values should be shown more consistently across datasets within figures.

Version 1:

Reviewer comments:

Reviewer #1

(Remarks to the Author)

The revised manuscript quality has been improved, and most of my concerns were addressed and resolved.

Reviewer #2

(Remarks to the Author)

The authors have addressed some of our points, but several major concerns remain. While I could move past most issues, point 1 is particularly important. It is crucial to demonstrate that the rescue is linked to mtDNA mutation load, rather than potential background mutations in the polGEXO- flies that are homozygous lethal. My previous comments outlined why addressing this is necessary. Additionally, multiple publications have shown that flies with high levels of mtDNA mutations are viable, including another polGEXO- fly model. The authors' response in the rebuttal letter, as well as the additional experiment involving five generations of polGEXO- heterozygous crosses, do not resolve this concern. It remains important to validate the effect of reduced autophagy in mitigating organismal defects caused by high mtDNA mutation levels in another fly model with high mtDNA mutations."

Reviewer #3

(Remarks to the Author)

The authors have addressed my concerns sufficiently.

Version 2:

Reviewer comments:

Reviewer #2

(Remarks to the Author)

After reviewing the authors' responses, we still think that testing the rescue effect—at least for one candidate—in an additional model with high mtDNA mutation levels could enhance the manuscript, given the limited mechanistic insights provided by the current form. However, we appreciate the effort that has already gone into addressing the reviewers' comments, and the other two reviewers are satisfied with the current version. With these points in mind, we are happy to leave the final decision to the editorial board regarding the acceptance of the manuscript in its present form.

Reviewer #4

(Remarks to the Author)

The present study explores mechanisms that may ameliorate the negative effects of mutations in mtDNA. The authors perform a genetic screen using a mutator *Drosophila* line with a mtDNA -polymerase (Polgexo-) that is not able to proof-read efficiently, leading to high mutation loads in mtDNA, and consequently impaired physiological functions. The screen identified several pathways that reduce these negative effects, and the authors focused specifically on autophagy, which when reduced, was able to rescue the lethal phenotype of Polgexo-.

My review arrives at very advanced stage of peer-review - the paper has been through at least two rounds of review by three reviewers. Two of them are satisfied with the extent of the changes made thus far by reviewers. One reviewer, remains concerned that the rescued phenotype observed by the authors may not be directly attributable to mitochondrial mutation load, but may reflect other background mutations in the nuclear genome.

I understand the reviewer's concern - disentangling mito-nuclear effects is often challenging, even in the best genetically tractable systems like *Drosophila*. However, I feel the authors have been extremely thorough and diligent in providing string and abundant evidence that the effect is related to the mtDNA.

In the extensive, comprehensive, and - in my opinion - extremely patient - rebuttal they clearly address several points that strengthen their conclusions.

Of particular strength, the authors have carried out additional experimental validation to address the reviewer's concerns, which I find very convincing:

"we backcrossed the POLGEXO- allele repeatedly via the female germline, increasing the mtDNA mutation burden. Such flies, could no longer be rescued via our rescue genes. However, reintroduction of "fresh" mitochondria, restored rescuability. This experiment clearly establishes causality of the mtDNA mutation burden. It is not clear to us how a potentially lethal background mutation could be effective only when inherited via the female germline with a high mtDNA mutation burden."

There are many other lines of evidence detailed in the author's rebuttal - there's no point in listing them here again - but the level of evidence in support of the authors' conclusions is extremely strong.

REVIEWER COMMENTS

We would like to thank all reviewers for their helpful comments. We have made sincere efforts to perform additional experiments and respond to all the concerns raised.

A common topic among the reviewers seems to be how the genetic screens and crosses were performed and what material was used for analysis. The crosses behind the screen and behind generating samples from rescued mutator fly lines require 5 consecutive crosses, resulting in only a few genotypically correct individuals. Thus, the samples generated are finite, and every time we need new material, these crosses need to be reinitiated from the start. Additionally, the genetic screen was performed 3 independent times, starting with the same large deficiency lines but using different balancers to ensure that the balancers did not interfere with the screen. In all cases, the same targets were identified. We hope our efforts are appreciated as we consider that this manuscript provides new insights into potential pathogenic mechanisms and a plethora of useful resources and ideas for other researchers.

We genuinely appreciate the reviewers' feedback, their help in strengthening our conclusions, and their comments on improving the manuscript for the interested readership.

Reviewer #1 (Remarks to the Author):

In the manuscript titled " Preventing excessive autophagy protects from the pathology of mtDNA mutations " Najla El Fissi et.al demonstrate that some critical pathways including nutrient sensing, insulin signalling, mitochondrial protein import, and autophagy that can rescue from the lethal phenotype induced by mtDNA mutations. The findings presented in this manuscript are potentially interesting to the readership, however, some data do not sufficiently support the conclusion.

We thank the reviewer for taking the time to review our manuscript.

Major comments

1. In Figure 4A, an increase in lysosomes (as indicated by lysotracker staining) was observed in mtDNA mutator larvae. To fully assess the integrity and activity of lysosomes, additional methods should be employed in both the mtDNA mutator larvae and the rescue lines.

Our initial submission included measuring the number of lysosomal foci, ATG8 translocation upon BafA1 treatment, and ATG8 lipidation. These together suggest an increased autophagic flux in mutator larvae, which normalised in the rescued larvae.

We also attempted to use reporters such as *uas::mCherry-GFP-Atg8a* (BL: 37749) or *uas::Mito-QC* (BL: 91641) reporter strains, but it became technically impossible to generate these animals. This strategy requires us to combine 5 different genetic elements, which turned out to be too toxic on the mutator background, and we did not generate any animals for analysis.

Therefore, we measured lysosomal activity via cathepsin-B activity (MagicRed), which is now presented in the manuscript. Neither LysoTracker nor MagicRed suggests any obvious changes in lysosome morphology.

2. Protein levels may not precisely reflect mitochondrial activity; hence, it is advisable to measure ATP production, mitochondrial ROS, and mitochondrial membrane potential in the mtDNA mutator larvae and rescue lines to obtain a more accurate assessment of mitochondrial function.

Protein levels were only used to investigate mitochondrial mass. Mitochondrial activity was tested by measuring respiratory chain enzyme activities directly from isolated mitochondria in all larvae models. We presented respiratory chain enzyme activities for complex I, complex I+III, complex II, complex II+III, and complex IV and have now added ATP synthase (complex V) activity.

Mitochondrial respiration was already shown to be unchanged in the D263A mtDNA mutator model in our previous work (Bratcic et al. (2015) Nat Comm). We, therefore, see no reason to reanalyse these. Measuring cellular ATP levels is not necessarily informative, as cells will attempt to maintain a usable ATP/ADP ratio at all costs.

We now present aconitase activity to reflect mitochondrial ROS levels.

We now present mitochondrial membrane potential, as determined by TRME staining.

3. In Extended Data Figure 9C, the ratio of Atg8a-I/Atg8a-II should be measured to assess the activity of autophagy.

According to the *"Guidelines for the use and interpretation of assays for monitoring autophagy"* (4th edition) (Klionsky et al. (2021) Autophagy 17(1)) the conversion of non-lipidated to lipidated Atg8a reflects the autophagic flux, not the autophagic activity. Furthermore, according to these guidelines, one should measure the Atg8a-II levels in the presence or absence of a lysosomal inhibitor such as BafA1 in relation to a housekeeping gene. We provided this in Extended Figures 9C and D, now Extended Figures 12f and 12g.

4. The mitochondrial morphology could be analyzed by confocal microscopy or transmission electron microscopy in the mtDNA mutator larvae and rescue lines.

As mentioned above, we used an in vivo approach to investigate mitochondrial morphology, using the reporter *uas::mCherry.mito.OMM* (BL: 66533). However, due to the number of transgenes required, it became technically impossible to generate the desired animals.

We, therefore, used confocal images from ATPase staining of BafA1-treated fat tissue to address this question. Blocking mitochondrial degradation was necessary due to the severely disturbed mitochondrial network in the mutator larvae. These data are now presented in Extended Figure 11.

5. It is interesting yet puzzling that only the nucleation, but not other processes of autophagy rescue the mutator lethality, it should be explained and discussed in detailed.

We agree with the reviewer that this observation is fascinating. We have further experimental evidence using autophagy inhibitors acting at the initiation or nucleation stages, corroborating our results. While inhibiting autophagy initiation also rescued the mutator larvae, blocking nucleation was far more effective. It is thus possible that the mere deletion of a single copy of an autophagy initiation factor is insufficient for rescue. This new data has been added to the manuscript in Figure 6b and further discussed.

Reviewer #2 (Remarks to the Author):

In this paper, Fissi et al. utilized a *Drosophila* model to identify genes that help counteract the developmental lethality caused by an mtDNA polymerase mutant D263A that is defective in proofreading. By conducting a deletion screen, they found removing one copy of several genes associated with nutrient sensing, insulin signaling, mitochondrial protein import, and autophagy that rescued the lethality of homozygous D263A. They then performed proteomics to profile the difference between D263A with and without the hemizygote mutation of the rescue genes, tested a few other autophagy components, and concluded that the reducing autophagy, particularly the components involved in the nucleation step, attenuated the impact of accumulated mtDNA mutations in the D263A mutant.

This work harnesses the power of *Drosophila* as a model system for genetic screening. Various techniques including proteomics and imaging were adapted to support their findings. However, the key question “how and why reducing autophagy nucleation alleviates the developmental lethality linked to D263A mutants and its relation to mtDNA mutations” remains unaddressed. Many data lacked proper explanations or speculations, which made the manuscript challenging to read and comprehend. Furthermore, the lack of key information in the figure legends and methods section, such as the specific genotypes and samples used for each experiment, makes it difficult to interpret the results.

While the deletion mapping is neat and the omics data could be valuable for other researchers, some of the major conclusions need further validation and certain observations would benefit from additional investigations. One would expect to gain more understanding of the topic from a manuscript in *Nature Communication*. Below I list some of my concerns.

We are sorry that the reviewer had this experience. However, it is important to point out that the key question was not “how and why reducing autophagy nucleation alleviates the developmental lethality linked to D263A mutants”. Rather, the key question of this study was to investigate if “...the progressive mitochondrial dysfunction observed in these mtDNA mutator models provides the opportunity to identify mechanisms that enhance tolerance to mtDNA mutations, reduce the mutation burden, or compensate for the OXPHOS dysfunction....” (line 59–61 in the main text).

Nevertheless, we believe we did address why the reduction of autophagy can benefit the organism. Our results suggest that an increased mitochondrial turnover is too energetically demanding for an already fragile organism, thus resulting in the developmental lethality observed here. Our results demonstrate that excessive autophagy has catastrophic consequences, and preventing this can stabilise mitochondrial mass, albeit these mitochondria are not fully healthy. It is well established that increasing mitochondrial mass is a common cellular mechanism in response to a mitochondrial defect. Why increased autophagy is initiated rather

than increasing mitochondrial mass in the mutator larvae is not clear and will require further work, but the mutation burden might play a role in this.

We would also like to point out that we did not “delete” autophagy. The heterozygous deletion only prevented an excessive activation. Thus, the identified factors might be important in the cellular response to mitochondrial dysfunction and the associated activation of autophagy. We now present data demonstrating that we can target autophagy pharmacologically, strengthening our results.

We hope that we clarified some of the reviewer's issues and that the additional experiments and modified manuscript have improved their impression of our work.

Major points:

1. The whole manuscript is based on the developmental lethality phenotype of the homozygote D263A mutant used for this study. However, whether the lethality results from the accumulation of mtDNA mutation was not addressed. This is especially important because there are other fly mutator models showing this is not the case. For example, Samstag et al (PLoS Genetics, 2018) showed that D263A/polG deficiency flies accumulated mtDNA mutations, yet >60% of them survived to adulthood. Another fly mutator model mito-APOBEC (Andreazza et al, Nature Communication, 2018) also showed ubiquitous expression of mito-APOBEC from early development is highly mutagenic for mtDNA, but flies reach adulthood although they have reduced lifespan. Interestingly, although APOBEC1 flies carry a high level of mtDNA mutations, they don't show a reduced mtDNA copy number. Additionally, it is known that *Drosophila* can tolerate high levels of mtDNA with large deletion (e.g. Lingenhohl et al, PNAS 1992). These made me wonder whether the developmental lethality observed with the D263A homozygous mutant used for the current manuscript is caused by the polGEXO- itself, a higher level of mtDNA mutations carried by this D263A mutant compared to other mutator models, or due to other factors in the genetic background. Hence, it is important to test the effect of four key genes (*atg2*, *tim14*, *melt* and *dilp1*) in a different mutator model to show whether the rescue is true in flies carrying high levels of mtDNA mutations, independent of how the mutations were introduced. The authors could for example express mito-APOBEC1 and examine whether removing one copy of *atg2*, *tim14*, *melt* or *dilp1* rescued the defects associated with mito-APOBEC1 expression. This experiment will strengthen the manuscript's focus on the pathology of mtDNA mutations rather than on POLG1, as indicated by the title.

We thank the reviewer for this interesting discussion. However, we are not sure we fully understand the question. The mtDNA mutator model used here expresses a proof-reading deficient mtDNA polymerase *Polg^{exo-}* from the endogenous *tamas* locus. We previously generated this model by ends-out homologous recombination and the resulting animals differ only in the D263A mutation in POLG1. The model was previously studied by us and others (for instance: Bratic et al. (2015) Nat Comm, Kauppila et al. (2018) PNAS, and Andreazza et al. (2019) Nat Comm), and shown to accumulate random mutations in mtDNA consisting of point mutations and indels. Homozygous *Polg^{exo-}* larvae and mice also present with reduced mtDNA copy number. The function and consequences of the *Polg^{exo-}* allele has further been studied in vitro, in yeast, worms, and mice. It is thus reasonable to conclude that the observed phenotype is caused by the consequences of the lack of exonuclease activity.

The reviewer is correct that others have reported different mtDNA mutator models that survive to adulthood. However, we would like to point out that these models have major differences that most likely explain the discrepancy.

Lingenhöhl et al (1992) PNAS reported a fly strain with large mtDNA deletions. A follow-up study showed that this strain had increased mtDNA levels compared to the controls (Beziat et al. (1993) NAR). This agrees with our work, where we present increased mtDNA levels in the rescued lines as part of the rescue mechanism.

Samstag et al expressed a D263A *Polg^{exo-}* mutant transgene in the background of a deletion strain (Df(2L)BSC252), removing the wildtype *tamas* locus. However, Df(2L)BSC252 does not only affect the *tamas* locus but also close to two dozen other genes, several of which directly or indirectly affect mitochondrial function. For instance, not only are POLG1 and POLG2 deleted in Df(2L)BSC252, but also CG33649, the mitochondrial glutamyl-tRNA amidotransferase, and mRpS23, a subunit of the mitochondrial ribosome. It is plausible that the Df(2L)BSC252 strain has a compensatory upregulation of mitochondrial biogenesis to compensate for the loss of these loci. To our knowledge, Samstag et al. did not investigate mitochondrial biogenesis or mtDNA copy number in their lines. Thus, both mentioned models demonstrated or have a likely upregulation of mitochondrial biogenesis, which is also part of the rescue mechanism reported in our study.

Finally, Andreatza et al. (2019) Nat Comm expressed a mitochondrially targeted cytidine deaminase (APOBEC1). This model accumulated C>T transitions, leading to an OXPHOS deficiency and reduced lifespan. The authors only presented results from adult flies, comparing them to age-matched heterozygous *Polg^{exo-}* flies. In the manuscript presented here, we used homozygous *Polg^{exo-}* larvae, and it is not clear what the mutation load of APOBEC1 larvae is. Furthermore, no reduction in mtDNA copy number or the accumulation of indels was reported. Thus, the two models differ substantially, and the authors state that the mutation spectra rather than the overall mutation burden correlate with the organismal fitness. This agrees with the results presented here, where the rescue via *tim14*, *atg2*, and *dilp1* did not affect the mutation burden. We would also like to point out that, to our knowledge, the methods used to calculate the mutation burden differ in the above reports, making direct comparisons difficult. Most importantly, Andreatza et al. report the total number of mutations, including multiple occurrences of the same mutation, while the original *Polg^{exo-}* report by Bratic et al. (2015) reported only unique mutations. Given these observations one should be cautious when comparing mtDNA mutation loads between these reports.

Nevertheless, to address the reviewer's question of whether the mtDNA load is responsible for the lethality, we intercrossed heterozygous *Polg^{exo-}* flies for up to 5 generations and demonstrate that while the mutation burden increases with consecutive interbreeding, only the F1 generation of homozygous mutator flies are rescued by the identified mechanism. These results clearly link the mutational burden to the rescue potential and are now presented in the revised manuscript.

2. The manuscript showed an interesting observation that only the nucleation step of autophagy is crucial for the rescue. However, it falls short of addressing why upstream or downstream of the same pathway does not elicit the same effect. It is a selling point for the manuscript, and one would expect authors to provide more insights/explanations for such discrepancies. For example, it raises questions about whether the nucleation step triggers additional processes besides autophagy, or if the nucleation components are rate-limiting factors of the entire autophagy process, so removing one copy of these factors are more likely to produce a phenotype than components in other autophagy steps. All experiments conducted involved deletion or RNAi. It would be interesting to do gain-of-function experiments to test whether upregulating autophagy is sufficient to reduce mitochondrial DNA copy number, and mitochondrial mass and cause the lethality from the pupa to adult transition in the absence of the mutator allele.

We agree with the reviewer that the observation of nucleation being the only targetable step is interesting – we would even dare to say “remarkable”. We believe that the generation of gain-of-function mutants to study autophagy is beyond this manuscript. We already tested 17 factors directly involved in autophagy and factors involved in mTOR signalling, insulin signalling, and mitophagy.

Instead, we used pharmacological inhibitors of autophagy initiation and nucleation to address this question. While inhibiting nucleation had a greater effect, impeding initiation also improved the mutator phenotype. We conclude that while nucleation has the broadest rescue potential, other steps of autophagy can also be effectors. However, their action might have different gene dosage responses and thus were not identified in our screen. It should be remembered all we did was reduce the gene copy number of one gene. These new data are now part of the revised manuscript.

The question of whether autophagy activation contributes to the phenotype remains unanswered. We want to point out that we are not reporting that mutations in the autophagic process led to disease. This has been demonstrated previously. Rather surprisingly, we show that under the right circumstances, autophagy is activated to such a degree that it becomes a driving force of the phenotype.

To explore this further, we now demonstrate that homozygous *Polg^{exo}*- larvae grown in the presence of the mTOR inhibitor rapamycin exhibit an aggravated lethal phenotype. This further supports our conclusion that excessive autophagy is part of the pathology of the mtDNA mutator model. These new data are now part of the revised manuscript.

3. It is not clear whether deficiency + D263A or mutant allele of the genes of interest + D263A were used for all phenotyping, proteomic analysis and imaging presented in Fig 2–4. If deficiency lines were used, the difference reported could stem from other genes deleted in the same def lines used. If p-element mutants were used, the authors need to explicitly state the lines used in the main text and the corresponding figure legends. Additionally, the validation of the mutants used as true KO mutants is lacking (e.g. 17156 has the p-element insertion in the 5' UTR of *Atg2* and does not seem to reduce the mRNA level of *Atg2* as shown in Fig S2C). Furthermore, it is not shown whether the proteomic and lysosome changes observed with hemizygotes (Fig 2 and 4) are linked to mtDNA mutation levels. In other words, the authors should present the mass spec proteomic data of hemizygotes in the absence of the D263A mutation, at least for some genes.

We apologise that this was not clear to this reviewer.

Deficiency lines were only used during the screening. All subsequent experiments used mutant lines. Everything else would make no scientific sense, as multiple genes would be affected in a deficiency line, and it would have been a waste of time to identify the responsible gene in the first place. We have now added a clearer description of this. Thus, the lines investigated were either homozygous for the *Polg^{exo-}* allele or homozygous for the *Polg^{exo-}* allele and heterozygous for one of the rescue genes. Heterozygosity was either due to a point mutation in the gene of interest or due to a P element insertion.

We thank the reviewer for their observation regarding *atg2* expression levels. We indeed measured expression levels of the individual rescue genes in *tamas* control, *wDah* control, heterozygous p-element insertion mutant, small deficiency lines, mutator, and their respective rescue lines (Previous Extended Figure 2C). However, we now noticed that we were unlucky with our labelling, resulting in the rescue-mutator and rescue-mutant receiving the same shortening (rescue-mut). We have corrected this in the updated extended data figure 2. We also split the figure into two and clarified it further in the figure legend. Nevertheless, the *atg2* expression levels are indeed reduced relative to a wildtype control, and the 3x increase observed in the mutator lines is severely blunted in the rescued lines. We hope that this is now clearer in the revised manuscript.

Although we did not intend to study the function of the identified rescue genes on their own in detail, we nevertheless now report the proteomic profile of heterozygous and homozygous mutant lines as obtained from Bloomington, except for *atg2*, where the homozygous genotype is embryonic lethal. We hope this will be useful to the community.

4. Did four rescue lines rescue the D263A in the same manner? The authors have reported their differences in various assays (mtDNA mutation levels, PGC-1alpha expression etc), but surprisingly, they all seem to down-regulate autophagy/lysosome. I found this puzzling, and the authors do not provide proper explanations or hypotheses.

We apologise if the reviewer felt we were not clear in our writing. We have now added to the discussion.

Minor points:

1. The mtDNA mutation level in Fig 1I should be presented in actual values, not in relatives (refer to the major point 1 to see why such information is needed). This also applies to Fig 1H and J.

The mutation frequency (Fig1I) was measured using the RMC method, which usually returns a ratio. It is possible to report a frequency, but we do not favour this way. Instead, we now additionally report the mutation frequency after cloning and sequencing a region covering ATPase6 and COXIII in control, *mutator*, *melt-mut*, and *atg2-mut*. We sequenced ~150–220K nt from ~180–268 clones of each of these lines. The results are now shown in the manuscript as Figure 2c.

It is unclear what the reviewer requests regarding Figures 1H and J. Does the reviewer request the absolute quantification of protein (Fig. 1H) or mtDNA (Fig. 1J)

levels? This seems a bit obsolete since we are comparing different experimental groups to control group, and the additional knowledge of knowing exact numbers of specific proteins or the mtDNA in a tissue or a cell is questionable. For instance, to get absolute number of a specific protein we would need to perform targeted mass spectrometry for a number of proteins and use heavy internal standards for each protein/peptide). We respectfully disagree with such an unreasonable request, and we are using methods that are state-of-the art and accepted by the scientific community in general.

Presenting the data in Fig. 1H and Fig. 1J in any other way but normalised will make comparisons between the different genotypes difficult. The raw values of these measurements can be found in the provided source data file.

2. It would be nice to state what assigned color labels on the confocal images directly to help read Fig 3B, 4A and 4D.

We have added the description.

3. The images presented in Fig 3B and 4A missing key information: which part of gut/CNS and what cell types are in the representative image? This is essential information as cells in different parts of the gut/brain are very different. Are all the images in those panels from the same cell type and tissue part? If yes, please show markers to confirm this. An alternative approach is to make genetic clones to show the differences in neighboring cells.

Of course, we investigated the same anatomical regions. We now present images identifying the investigated anatomical region clearly. The middle midgut can be identified by the unique morphology of the cells in this region, images to show this have been added to the manuscript.

4. What is the difference between *wDah* and *tamas*, and which one should be considered as the control? Why were *wDah* data only shown in Fig2 and 3A, but not Fig 1 and 4? This is important since 3A showed *wDah* had similar mito protein content as mutator but *tamas* control did not.

WhiteDahomey (*wDah*) is the genetic background of the *Polg^{exo-}* mutator and *tamas* control line. As discussed, the *Polg^{exo-}* model was generated by ends-out homologous recombination, where the *POLG* locus (*tamas*) was first replaced by a short attP site, which is then targeted for the reintroduction of genetic information. The *tamas* line in this manuscript represents the reintroduction of the wildtype genomic *tamas* allele, while the mutator (*Polg^{exo-}*) represents the reintroduction of genomic *tamas* with the D263A mutation. Thus, these two lines are the preferred controls, as they underwent the same genetic manipulation and are maintained with the same balancers. The proteomic data provided in this manuscript suggest that the *wDah* and *tamas* lines, indeed, are very similar, and mild differences might be a reflection of the different strain maintenance methods.

5. The method section should contain details describing how different complex activities were measured (Fig 1F).

The mitochondrial respiratory chain complex activities of complexes I to IV were measured in our clinic (CMMS) at the Karolinska University hospital, using the same method used to diagnose patients with suspected mitochondrial disease. The reference we refer to was written by staff at our clinic and describes the exact

method still used today, and which was applied to analyse the fly samples. We and others have used this method for numerous mouse and fly models for the last two decades or so. We have added a short summary here, but this cannot be as detailed as the description of the original methods paper. Complex V activity, which we have added to this revised manuscript, is new to the clinic's repertoire and we therefore provide a detailed description here.

6. Fig 2D and 3E, please label the significantly up/down regulated protein as 4C and mark PGC1a in 3E since they mentioned it in the main text.

Labels are now added. Figure 3E (now 4e) shows protein levels, while the text refers to PGC1a transcript levels (now Extended Data Figure 12a).

7. Fig 3A what is the $-\log_{10}$ and ratio rel

The $-\log_{10}$ label is an orphan label that we unfortunately forgot to remove during figure preparation. It is now removed. The 'ratio rel' is the protein expression ratio of mean of all mitochondrial proteins in a given sample and genotype over *tamas*. It could also be seen as the "intensity ratio of mitochondrial proteins relative to *tamas*". We have clarified this in the text.

8. Fig 4D, why fat tissue? Why were CNS and gut cells chosen for Fig 4A and 3B?

The mentioned tissues were chosen for their suitability for the applied techniques and clarity of the images.

9. The nature of D263A/*tamas* control and how they were generated should be provided in the main text or the method section. Citing the Bratic et al (2015) paper, which contains a detailed description of how the D263A mutant was generated, is not sufficient. A brief description is needed in this manuscript, so the readers don't need to refer to other publications to understand the nature of the mutant.

Text has been added.

10. Fig S4A, B, are they mtDNA copy numbers or mutation loads? The figure legend does not seem to match.

We apologise for this mistake. Figure S4B (now Extended Data Figure 5b) represents mtDNA levels. We have corrected the label.

11. There is no data to support the statement on the melted rescue (line 113) being "early in the fly development". In fact, according to Flybase, the expression of melted continued to be low and only upregulated to moderate expression during the pupae stage. The author should provide a proper explanation or data to support this statement.

We would like to point out that while we do not provide data for this statement, we did not make this claim either. The sentence reads "In contrast, *melt*-mut larvae kept a mutation burden comparable to controls (...), suggesting that *melted* may act early in fly development before the accumulation of mtDNA mutations." This sentence clearly is a discussion point and the wording used shows this. We later also present that protein levels of BNIP3, shown to act early in development, are reduced in the *melt*-mut line, supporting our hypothesis. However, it is clear that we merely suggest an early involvement.

There are two possibilities for the reduced mutation burden. Either the mutations are generated but removed before the analysis time point, or they are never generated in the first place. To address this, we measured the mutation burden in control, mutator, and *melt*-mut flies at developmental stages S3, S11, and S13 (Figure 1 of this letter). Interestingly, mutator embryos have a fairly high mutational burden at S3 and S11, dramatically decreasing by S13. This behaviour is mirrored in the *melt*-mut embryos, albeit to a lower extent. These observations are compatible with the recent report of a BNIP3-dependent mitochondrial purging during fly development (Palozzi et al. (2022) Cell Metabolism). However, this data does not support either possibility; rather, it is a combination of both. Thus, much more work will be required to clarify this observation. For this reason, we prefer only to present this data here and not include it in the manuscript. Finally, our cloning and sequencing data also points to *melted* acting early, but more work is needed.

Figure 1. Relative mtDNA mutation frequency by RMC method. Embryos were picked 2h, 6h, and 13h after egg laying and used for random mutation capture analysis.

12. The melt rescue flies have much-reduced mtDNA mutation levels compared to other rescuers and D263A alone. Does this line live longer than the other rescuers after eclosure?

This is an interesting question, but we have not investigated the adult flies apart from their climbing abilities. We do know that all rescued lines survive for at least 10 days, with no apparent phenotype other than jumping. However, for this manuscript, we do not intend to report on the adult flies, as this would distract from the current volume of work.

Reviewer #3 (Remarks to the Author):

In their manuscript entitled "Preventing excessive autophagy protects from the pathology of mtDNA mutations", Fissi et al. study the consequences of mitochondrial DNA mutations resulting from mutant mtDNA polymerase in drosophila. The authors carried out a genetic screen in isogenic hemizygous deficiency strains to identify genes that can rescue the mtDNA mutator fly phenotype. They identified nine genes, part of autophagosome formation, mitochondrial protein import, insulin-like growth signaling, and nutrient sensing. To understand the courses underlying rescue, the authors analyzed by mass spectrometry the proteomes of different fly tissues with and without rescue. Lastly, the authors monitor specific pathways to observe reduced mitochondrial mass and increase macroautophagy in mtDNA mutator larvae.

The manuscript is well written and the data are of high quality. The manuscript builds on established technologies to identify rescue genes and describe resulting proteome changes. However, additional validation or mechanistic study of observations is necessary to extract additional new insight.

We thank the reviewer for their time assessing our manuscript.

Main comments:

- 1) The overall role of autophagy in protecting from mtDNA mutations remains unclear. Ext Data Fig 9 C, D: It is unclear what was quantified, are the shown values ratios of the lipidated versus non-lipidated Atg8 and or some comparison between DMSO versus BafA? The overall experiment is difficult to interpret and a more clear description of the method and interpretation would be helpful. This includes the vastly different effects of the tamas controls across in the two Western blots. In addition, how can the observed Atg8 degradation in tamas upon BafA treatment be explained? This appears counterintuitive. Overall, the data does not appear to be strong enough to support the main claim of the title that preventing excessive autophagy is protective for mtDNA mutations. Additional experiments are required to show the autophagy effects and particularly to support the point of "excessive" autophagy.

We have now clarified the figures (now Extended figure 12f and g. These figures quantify the autophagic flux, according to the *"Guidelines for the use and interpretation of assays for monitoring autophagy"* (4th edition) (Klionsky et al. (2021) *Autophagy* 17(1)). Please also see our response to reviewer 1. We measure the amount of Atg8a-II in relation to a housekeeping gene in the presence or absence of BafA1 (DMSO is used as a control). Differences in intensity on the gel stem from unequal loading of the shown gel, but each experiment was performed 3 independent times, and the average is shown in extended figure 12g.

Additionally, our conclusions are supported by immunohistochemistry and the additional experiments detailed to reviewer 1 point 1.

- 2) The lack of identifying Parkin in the screen is not sufficient to rule out a role for mitophagy in the observed reduction in mitochondrial mass. Additional experiments that directly target mitophagy (e.g. via Pink1, Parkin) are required to make this point.

We apologise for the confusion. We directly tested PINK and PARKIN using gene-specific mutants. Neither line could rescue the mutator phenotype. We have now clarified this in the text.

- 3) The identified rescue genes are a key outcome of the manuscript. Strikingly, most genes hit the same pathways, strongly supporting these as correct. Considering the important impact of

the remaining genes of the import and nutrient sensing pathway to the manuscript, additional validation experiments depleting other components of these pathways would be important.

In our initial submission we used 17 genes of autophagy and 9 genes of insulin signalling. We already thought that this was more than enough, considering the extent of the initial genetic screen. We now additionally tested TOM40 of the protein import machinery and 7 genes involved in TOR signalling. None of these targets were able to rescue the mutator. Dissecting each one of these pathways further should be left for future work.

4) The proteomics data is an important asset of the manuscript. Please make the data more accessible by providing full processed data as supplementary table.

We now provide additional information regarding the proteomic data as Extended Data Tables 3 and 4. As stated in the Data availability section of the manuscript, the proteomic data had already been available at the ProteomeXchange Consortium.

As requested by reviewer 2, we now also provide the proteomic profiles from heterozygous and homozygous mutants for *atg2*, *tim14*, *dilp1*, and *melted*.

Minor comments:

5) It would be helpful to the reader to have an explanation for why the 3rd chromosome specifically was screened.

We have now provided additional information. As *Drosophila* only has 4 chromosomes, and *tamas* is encoded on chromosome 2, chromosome 3 was the largest remaining chromosome to screen.

6) Are the effects observed in Fig 3A,D statistically significant?

For Figure 3A (now 4a), the reduced mitochondrial mass in mutator versus *melt-rescue* and *wDah* versus *melt-rescue* are significant, all other genotypes are not significant. For Figure 3D (now 4d), BNIP3 levels are only significantly changed in the *melt-rescue* line.

7) Fig 1F: y-axis description is missing

Thank you for noticing. It has been added.

8) Fig 1D,H: points should be used rather than commas in numbers (e.g. "1.5" rather than "1,5"). This also occurs in other figures.

Continental Europe routinely uses commas instead of points. We have now changed our figures to points but leave it to the editor to advise us on which punctuation the Nature publishing group prefers.

9) Fig. 2D, Fig. 3E: it would be helpful to indicate the base of the logFC shown (i.e. log₂FC).

Has been added

10) Fig. 3A: There is a misplaced textbox "-log₁₀(adj.P.Val)" at the bottom of the panel.

Thank you for noticing. We have removed the text box.

11) Many figures only have very selected statistical information indicated and p-values should be shown more consistently across datasets within figures.

Statistical significance is indicated on all graphs on all figures except the volcano plots, where statistical significance was not obviously shown. We have amended this. We hope this is what this reviewer was referring to.

We thank the reviewers for their assessment of our revised manuscript. Please find our comments regarding the outstanding questions below. Changes in the manuscript have been marked in red.

Reviewer #1 (Remarks to the Author):

The revised manuscript quality has been improved, and most of my concerns were addressed and resolved.

We thank the reviewer for taking the time to review our manuscript.

Reviewer #2 (Remarks to the Author):

The authors have addressed some of our points, but several major concerns remain. While I could move past most issues, point 1 is particularly important. It is crucial to demonstrate that the rescue is linked to mtDNA mutation load, rather than potential background mutations in the polGEXO- flies that are homozygous lethal. My previous comments outlined why addressing this is necessary. Additionally, multiple publications have shown that flies with high levels of mtDNA mutations are viable, including another polGEXO- fly model. The authors' response in the rebuttal letter, as well as the additional experiment involving five generations of polGEXO- heterozygous crosses, do not resolve this concern. It remains important to validate the effect of reduced autophagy in mitigating organismal defects caused by high mtDNA mutation levels in another fly model with high mtDNA mutations.

We value the reviewer's ongoing concern but are unsure why our prior response, along with the additional experiments, did not address it fully. We would be glad to further clarify our rationale and explain why we believe the proposed experiment may not directly address the reviewer's question.

The reviewer has raised concerns about the validity of our POLGEXO- fly model and requests additional validation using a different mtDNA mutator model. Their main argument is that published studies report adult flies with high mutational burdens, leading the reviewer to hypothesise that an unidentified background mutation may be responsible for the observed larval lethality in our homozygous POLGEXO- flies.

We respectfully note that this hypothesis does not align with our findings, and, upon careful review, we believe it is not supported by the available evidence. Additionally, the reviewer's comments do not fully account for the extensive validation and additional experiments we have conducted to support our model.

The reviewer cites three publications reporting adult flies with high mutation burdens in an effort to compare our homozygous POLGEXO- fly larvae to these models. However, we believe this comparison involves several critical misunderstandings:

1. The reviewer directly compares the mutations measured without accounting for their different types and quality.

2. The reviewer fails to account for the different techniques used to measure mutation frequencies, which vary significantly across studies.
3. The developmental stages at which measurements were taken are not considered, despite their importance in interpreting results.
4. Finally, the reviewer overlooks substantial differences in the mutation frequencies observed in these models.
5. Differences in the genetics of the different models was not considered

Below we clarify these points further to address any remaining concerns.

To point 1: Comparing different types of variants

The models brought forward by the reviewer either study a large deletion in mtDNA (Lingenhöhl et al. (1992)), random point mutations and single nucleotide deletions (Bratic et al. (2015) and Samstag et al. (2018)), or C:G>T:A transitions (Andreazza et al. (2019)). It is well established that different variants can have very different metabolic and cellular consequences, making direct comparisons impossible.

We already mentioned previously that the flies reported by Lingenhöhl present with increased mtDNA levels, explaining their survival.

The APOBEC1 model by Andreazza et al. introduces C:G>T:A transitions, while the POLGEXO-models present with an entirely different mutation spectra. Thus, differences in phenotypes can be attributed to these differences, making neither model right nor wrong, just different. Consequently, the reviewer deviates from our initial hypothesis, where “we postulated that progressive mitochondrial dysfunction [...] provides an opportunity to identify mechanisms that enhance tolerance to mtDNA mutations, reduce the mutation burden, or compensate for OXPHOS dysfunction.” Our work provides evidence for two of these mechanisms: The heterozygous deletion of *dilp1*, *tim14*, and *atg2* compensates for the mitochondrial dysfunction, while *melted* reduces the mutational burden. The reviewer’s requested experiment imposes an entirely new question onto the manuscript, which is better addressed separately.

To point 2: Comparing techniques

It is well established that mutation frequencies determined by different techniques are not directly comparable. Bratic, Samstag, and Andreazza use different approaches to measure mutation frequency in their models. In Bratic et al. and the current manuscript, we used cloning and sequencing, while Samstag and Andreazza used a duplex sequencing approach.

Andreazza and Bratic most clearly demonstrate this difference. Both groups determined the mutation frequency in adult heterozygous POLGEXO- flies. These flies are the same model and were provided by us to the Andreazza lab, allowing for comparisons of the methods used. Bratic et al. reported a mutation frequency of $\sim 1 \times 10^{-4}$, while Andreazza et al. measured a frequency of $\sim 7 \times 10^{-4}$. Thus, the two methods used to determine mutation frequencies are not

directly comparable, with duplex sequencing overestimating the mutation burden compared to cloning and sequencing.

With this in mind, Andrezza et al. report a mutation frequency of $\sim 6 \times 10^{-4}$ in their APOBEC1 flies, which is lower than the frequency the authors report for heterozygous POLGEXO- flies. Since homozygous POLGEXO- larvae present with a significantly higher mutation frequency than their heterozygous siblings, it is reasonable to assume that homozygous POLGEXO- larvae would have a significantly higher mutation burden using duplex sequencing.

Thus, by normalising for the different methods used, the differences in phenotypic presentation between these two lines is not surprising.

To point 3: Mutation frequencies at different developmental stages

We previously reported a burst in mtDNA mutations after morphogenesis in heterozygous POLGEXO- flies (Bratic et al. (2015)). This observation is crucial and must be considered in the context of our current findings. To our knowledge, neither Samstag et al. nor Andrezza et al. measured the mutation frequency in larvae in their models, while we do not report the mutation frequency in adult POLGEXO- rescue flies. Thus, mutation frequencies of the three models were never measured at the same developmental stages, making direct comparisons difficult. As mentioned above, the only comparison possible is between adult heterozygous POLGEXO- and APOBEC1 flies. Since these flies have similar mutation frequencies as adults, it is reasonable to assume that their mutation frequencies are similar also at the larvae stage. From this, it can only be concluded that APOBEC1 larvae must have a substantially lower mutation frequency as larvae than homozygous POLGEXO- larvae.

To point 4: Mutation frequencies

Importantly, even adult heterozygous POLGEXO- and APOBEC1 flies have a lower mutation frequency than homozygous POLGEXO- larvae (compare Andrezza et al. Figure 1a and Bratic et al. Figure 4). Therefore, it is reasonable to conclude that APOBEC1 larvae have mutation frequencies similar to heterozygous, not homozygous, POLGEXO- larvae.

Samstag et al. only report the mutation frequency in a highly genetically manipulated fly line that still expresses a wildtype copy of *tamas*. The mutation frequency in the Samstag adult POLGEXO- flies is significantly lower than that reported in our homozygous POLGEXO- larvae. We previously reported complementation between the different POLG molecules during replication (Bratic et al. 2015), fully explaining the lower mutation frequency reported by Samstag.

Given the points made above, it is inaccurate to suggest that other fly models have mutation frequencies similar to homozygous POLGEXO- flies, and the differences in survival can be attributed to the increased mutation load in homozygous POLGEXO- larvae.

To point 5: different models

Samstag et al. report adult homozygous POLGEXO- flies with a strong phenotype, prompting the authors to continue with POLGEXO- flies co-expressing a wildtype copy (see above).

However, this model differs substantially from the POLGEXO- model used by Bratic and us. Samstag et al. expressed a POLGEXO- polymerase from a large genomic construct, introduced into chromosome 3, while simultaneously deleting the endogenous *tamas* locus on chromosome 2, using two overlapping deficiency strains. However, while the genomic POLGEXO- construct also contains an additional 17 genes, the two overlapping deletions also remove seven additional genes and leave twelve more in a hemizygous state. Thus, the flies reported by Samstag et al. are highly manipulated, with the copy number and expression of numerous additional genes, including several involved in mitochondrial function and metabolism, being affected. The effects of these manipulations were not studied, but increased mitochondrial biogenesis is a reasonable assumption. This contrasts with the flies used in our study, where only a single amino acid substitution in *tamas* was performed.

Additional Points:

To address the reviewer's concerns, we demonstrate that the rescue potential segregates with the mtDNA mutation burden in the female germline. In this classical genetic experiment we backcrossed the POLGEXO- allele repeatedly via the female germline, increasing the mtDNA mutation burden. Such flies, could no longer be rescued via our rescue genes. However, reintroduction of "fresh" mitochondria, restored rescuability. This experiment clearly establishes causality of the mtDNA mutation burden. It is not clear to us how a potentially lethal background mutation could be effective only when inherited via the female germline with a high mtDNA mutation burden. We would appreciate further insights from the reviewer on how this experiment does not resolve their concern regarding causality of the mtDNA frequency.

Validation of the POLGEXO- Knockin Model:

The D263A knockin POLGEXO- mtDNA mutator model was extensively validated in the initial Bratic et al. publication. Since its creation 12 years ago, this model consistently failed to produce adult flies. The only exception being the experiments presented in the current manuscript.

The observations supporting the model are:

- The POLGEXO- flies were generated by homologous recombination, resulting in the modification of a single amino acid (D263A) only.
- The *tamas* control line used then and now underwent the same genetic manipulation and exhibited no phenotype.
- Deleting and reintroducing genomic constructs in the *tamas* locus did not affect the expression of surrounding genes.
- A developmental delay in heterozygous flies segregated with the mtDNA mutation load in the female germline but not in that of males.
- The specificity of the exonuclease deficiency was controlled for in compound heterozygous flies expressing the POLGEXO- allele and a polymerase-deficient but POLGEXO+ allele.

- The POLGEXO- flies have been maintained with continuous backcrossing to *wDah* for 12 years to exclude genetic drift. The lines were backcrossed before and during the preparation of this manuscript and showed consistent phenotypes and responses to the rescue allele.

Given this evidence, the reviewer's hypothesis of an unidentified background mutation of unknown function affecting POLGEXO- viability is speculative, as any unlinked variants would have been selected against during backcrossing, and linked variants would not perfectly segregate with the mtDNA mutation burden in the female germline.

In the current manuscript, we provide further evidence for our conclusions:

- The genetic screen was performed from the start three times with two different balancers and genetic backgrounds to exclude any cryptic genetic variation.
- We aggravated or mitigated the mutator phenotype by either increasing (via rapamycin treatment) or decreasing (via 3-methyladenosine) autophagy. Again, we fail to see why a potentially lethal background mutation would have precisely the expected phenotypic behaviour.
- All identified rescue genes can be directly or indirectly linked to autophagy, and the experiments performed in this manuscript directly demonstrate changes in autophagic flux and rescue of the OXPHOS defect.

In summary, while the reviewer raises questions about the validity of our model, we believe it is important to consider all the experiments that support our conclusions. Our results revealed that it is possible to rescue an OXPHOS deficiency by targeting the identified pathways. Finally, we would like to note that in the equivalent mtDNA mutator mouse model, homozygous POLGEXO- mice are not born at normal Mendelian ratios, suggesting that embryonic lethality is plausible (Trifunovic et al. 2004).

Reviewer #3 (Remarks to the Author):

The authors have addressed my concerns sufficiently.

We thank the reviewer for taking the time to review our manuscript.